# Distinct lateral inhibitory circuits drive parallel processing of sensory information in the mammalian olfactory bulb

Matthew A Geramita[1,2], Shawn D Burton[3,4], Nathan N Urban[1,2]*

[1]Department of Neurobiology, University of Pittsburgh, Pittsburgh, United States; [2]Center for the Neural Basis of Cognition, University of Pittsburgh, Pittsburgh, United States; [3]Department of Biological Sciences, Carnegie Mellon University, Pittsburgh, United States; [4]Center for the Neural Basis of Cognition, Carnegie Mellon University, Pittsburgh, United States

**Abstract** Splitting sensory information into parallel pathways is a common strategy in sensory systems. Yet, how circuits in these parallel pathways are composed to maintain or even enhance the encoding of specific stimulus features is poorly understood. Here, we have investigated the parallel pathways formed by mitral and tufted cells of the olfactory system in mice and characterized the emergence of feature selectivity in these cell types via distinct lateral inhibitory circuits. We find differences in activity-dependent lateral inhibition between mitral and tufted cells that likely reflect newly described differences in the activation of deep and superficial granule cells. Simulations show that these circuit-level differences allow mitral and tufted cells to best discriminate odors in separate concentration ranges, indicating that segregating information about different ranges of stimulus intensity may be an important function of these parallel sensory pathways.

*For correspondence: nurban@pitt.edu

Competing interests: The authors declare that no competing interests exist.

## Introduction

Brain sensory systems use parallel pathways to encode different components of sensory information. Motion and color are segregated in the visual system (*Merigan and Maunsell, 1993*; *Callaway, 2005*), sound location and tonal pattern are processed in parallel pathways in the auditory system (*Lomber and Malhotra, 2008*), and pain and itch are conveyed by distinct pathways in the somatosensory system (*Davidson and Giesler, 2010*; *Ross, 2011*). How local circuits support feature selectivity in these parallel streams remains poorly understood. This issue is of particular interest when considering brain areas in which distinct outputs are generated from initially homogeneous sources, such as in the olfactory system.

In the mammalian olfactory bulb, firing of olfactory sensory neurons (OSNs) excites two classes of projection neurons – mitral cells and tufted cells (MCs and TCs). While long viewed as essentially equivalent neuron classes, emerging evidence shows that MCs and TCs exhibit distinct activity in vivo (*Nagayama et al., 2004*; *Fukunaga et al., 2012*; *Igarashi et al., 2012*; *Adam et al., 2014*; *Otazu et al., 2015*), suggesting that MCs and TCs encode complementary aspects of olfactory information. For instance, TCs respond to lower odor concentrations than MCs (*Igarashi et al., 2012*; *Kikuta et al., 2013*), suggesting that TCs are involved in processing near-threshold stimuli. Consistent with this notion of parallel pathways, MCs and TCs project their axons to many non-overlapping regions (*Nagayama et al., 2010*; *Igarashi et al., 2012*). Recent work has begun to explore the circuit-level origins of these functional differences in odor-evoked activity. For instance, TCs are more intrinsically excitable and receive stronger OSN-mediated excitation than MCs (*Gire et al., 2012*;

**eLife digest** The brain often processes different features of sensory information in separate pathways. For example, when seeing an object, information about colour and movement are processed by separate types of neurons in the eye. These neurons in turn relay information to different sets of brain areas, all of which are active at the same time. Such parallel processing was originally not thought to apply to information about smell. This was because in mammals, the two types of neurons in the brain area that processes smell seemed to play the same role. However, more recent work suggests that there are in fact differences in the responses of these two neuron types (called mitral cells and tufted cells) to odors, suggesting that the brain might use parallel processing for information about smells too.

Information travels along neurons in the form of electrical signals, and this activity is often seen in the form of a series of "spikes". In a process called lateral inhibition, the activity of one neuron can feed back and inhibit the activity of its neighbors. This is important for enhancing contrast; in terms of the sense of smell, lateral inhibition is thought to help distinguish between similar odors.

A technique called optogenetics allows the activity of particular neurons in an animal's brain to be controlled by shining light onto them. Geramita et al. have now used this technique in mice to investigate whether there are differences in how lateral inhibition works in mitral cells and tufted cells. This revealed that lateral inhibition affects mitral cells only when they are spiking at intermediate firing rates, whereas tufted cells are only affected by lateral inhibition when spiking at low firing rates. Using computer simulations, Geramita et al. show that these different responses mean that mitral cells are best at distinguishing similar smells when they are present at high concentrations, while tufted cells are best at distinguishing similar smells that are present at low concentrations. These differences also mean that, by working together, mitral and tufted cells can distinguish between smells much better than either type of neuron on its own.

These results demonstrate that, as with the other senses, the brain processes information about smell using parallel pathways. Future work is now needed to see what effect switching off the activity of either mitral or tufted cells will have on an animal's behavior.

*Burton and Urban, 2014*). Whether other elements of the olfactory bulb circuit account for differences in MC vs. TC odor-evoked activity is unknown.

Throughout the brain, lateral inhibitory circuits enhance contrast and facilitate discrimination of similar stimuli by decorrelating neural responses (*Hirsch and Gilbert, 1991*; *Urban, 2002*; *Gschwend et al., 2015*). In the olfactory bulb, lateral inhibition occurs between pairs of MCs or TCs via reciprocal dendrodendritic synapses with inhibitory granule cells (GCs) (*Schoppa and Urban, 2003*; *Egger and Urban, 2006*). Previously, we have shown that lateral inhibition most strongly affects MCs firing at intermediate rates because coincident input is required for the activation of GCs (*Arevian et al., 2008*). This activity-dependent regulation of the strength of lateral inhibition onto MCs decorrelates MC responses to similar input more effectively than subtractive or divisive forms of inhibition (*Arevian et al., 2008*). Lateral inhibition onto TCs is largely unexplored, but the marked differences in MC and TC odor-evoked activity suggest that lateral inhibition onto TCs may operate differently.

Here, we show that optogenetic activation of a single, gene-targeted glomerulus elicits larger and more asynchronous lateral inhibitory currents in nearby MCs than in TCs. Moreover, this same photostimulation paradigm inhibits spiking differently in MCs and TCs. While MCs are affected by lateral inhibition at intermediate firing rates (~50 Hz), TCs are affected when firing at low firing rates (<25Hz). This difference arises, in part, due to differential recruitment of morphologically distinct classes of GCs by MCs and TCs. Finally, we use simulations to explore how these circuit-level differences between MCs and TCs influence odor discrimination. Specifically, the combination of activity-dependent lateral inhibition at both low and intermediate rates enables TCs and MCs to collectively encode odors better than either population alone and supports novel computations that are unlikely to occur with a single neuron type.

## Results

To analyze lateral inhibition, we optically activated M72-expressing OSN axons in acute slices from M72-ChR2-YFP mice (*Smear et al., 2013*) while recording from MCs or TCs innervating nearby glomeruli (*Figure 1a,b*). This approach allows specific and selective activation of a single, genetically-identified glomerulus across animals, eliminating an important potential source of variability. MCs and TCs showed reliable lateral inhibition following a 10 ms light pulse (*Figure 1c,d*), and similar proportions of MCs (10/17–59%) and TCs (9/15–60%) received lateral inhibitory currents. The MCs and TCs recorded were similar distances from the M72 glomerulus, and lateral inhibition did not vary within this limited range of distances in either MCs or TCs (*Figure 1—figure supplement 1*). To examine early and late components of the inhibitory responses onto MCs and TCs, we separated early (<250 ms after stimulation) and late (>250 ms after photostimulation) components of the inhibitory currents. The peak amplitude of early inhibition was larger in MCs than in TCs (*Figure 1e*), while the charge transferred of early phase inhibition was not significantly different (*Figure 1f*). Additionally, both the peak amplitude and charge transferred of late phase inhibition (>250 ms) was significantly larger in MCs than in TCs. MCs also receive a smaller proportion of total inhibition during the early phase than TCs, indicating that inhibition is more asynchronous onto MCs than onto TCs. Collectively, these results demonstrate that lateral inhibitory currents are larger and more asynchronous onto MCs than onto TCs.

We also explored two potential causes of cell-to-cell variability in lateral inhibition. The strength of lateral inhibition onto MCs and TCs did not depend on whether the apical dendrite was truncated during the slicing procedure, indicating that lateral inhibition originating in the glomerular layer (*Aungst et al., 2003*; *Liu et al., 2013*; *Whitesell et al., 2013*; *Banerjee et al., 2015*) did not significantly contribute to the differences in lateral inhibition studied here. Variability from slice-to-slice did contribute to variability in the strength of inhibition. However, for MCs and TCs recorded in the same slice, the same relationships in peak amplitude and charge transferred for both early and late phase inhibition were observed as in the larger data set. Specifically, early and late phase peak amplitude, but only late phase charge transferred, was significantly higher in MCs than in TCs measured in the same slice (*Figure 1g*).

To confirm that these results are not specific to the M72 glomerulus, we performed an analogous experiment in OMP-ChR2-YFP mice (*Smear et al., 2011*) by photostimulating a single unidentified glomerulus (100 ms pulses) in the medial olfactory bulb (*Figure 1h–i*). Previously we have shown that we can limit photostimulation to single glomeruli in OMP-ChR2-YFP mice (*Burton and Urban, 2015*). Similar to the results obtained using M72-ChR2-YFP mice, we find that only the peak amplitude of the early phase of inhibition is larger in MCs while both the amplitude and charge transferred of the late phase of inhibition are larger in MCs (*Figure 1j,k*). Together these two experiments indicate that lateral inhibition is larger and more asynchronous onto MCs than onto TCs.

Finally, using photostimulation of a single glomerulus in OMP-ChR2-YFP mice, we tested whether GCs contribute similar proportions of lateral inhibition onto MCs and TCs using a previously described strategy (*Najac et al., 2015*) to differentiate between GC- vs. non-GC-mediated inhibition onto MCs and TCs. We recorded lateral inhibition before and after limiting GC-mediated inhibition by bath applying NMDAR antagonist APV (25 μM) and mGluR antagonist LY36785 (100 μM). Using this pharmacological approach, we found that GCs contribute similar proportions of lateral inhibition onto MCs and TCs (*Figure 1l*). Additionally, in MCs and TCs in which the apical dendrite had been truncated, all lateral inhibition was blocked after limiting GC-mediated inhibition (*Figure 1i,l*). In contrast, small inhibitory currents remained after limiting GC-mediated inhibition in MCs and TCs with intact apical dendrites (*Figure 1i,l*). Removing cells with cut apical dendrites and redoing the analysis presented in *Figure 1l* similarly shows that GCs contribute similar proportions of lateral inhibition onto MCs and TCs. These observations indicate that glomerular layer circuits make small and uniform contributions to lateral inhibition in MCs and TCs and thus are not the primary source of differences in lateral inhibition between MCs and TCs.

We next explored how lateral inhibition influences MC and TC spiking. To do this, we stimulated MCs and TCs by step current injection and measured firing rates to construct input-output curves in cells near the M72 glomerulus. On interleaved trials we activated M72 OSN axons via photostimulation (*Figures 2,3*). We used a duration of step current injection matching the physiological duration

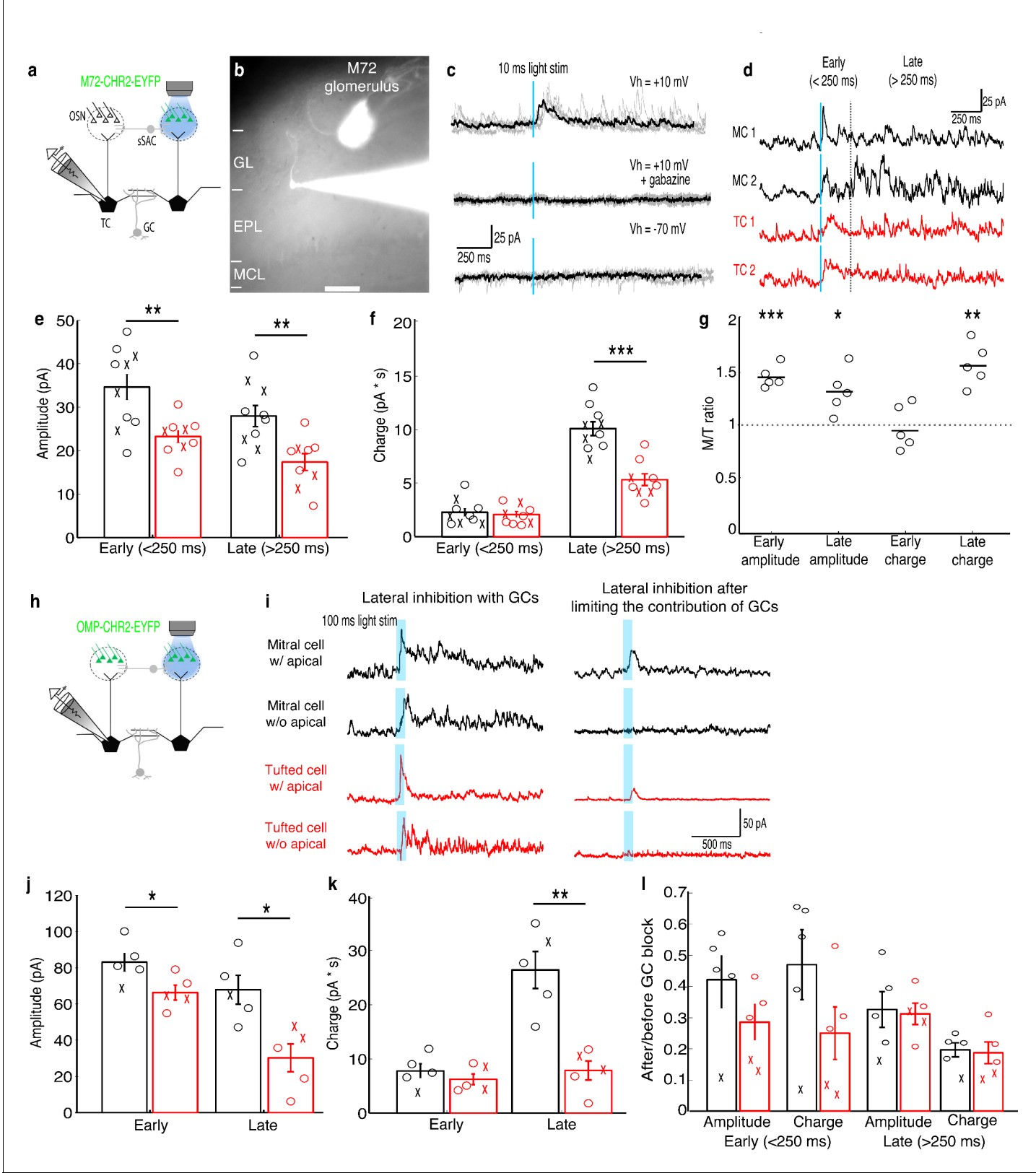

**Figure 1.** MCs receive stronger lateral inhibition than TCs. (a–b) Schematic and example of recording from a TC that projects to a glomerulus near the M72 glomerulus to measure lateral inhibition in M72-ChR2-YFP mice (GL: glomerular layer, EPL: external plexiform layer, MCL: mitral cell layer). Scale bar –100 μm. (c) Light stimulation evoked reliable inhibitory currents in recorded TC (TOP; 5 trials – grey, average – black) that are abolished by

*Figure 1 continued on next page*

*Figure 1 continued*

gabazine (MIDDLE) but did not evoke excitatory currents (BOTTOM). (d) Examples of average inhibitory currents in 2 MCs and 2 TCs. Inhibitory responses were grouped into early phase (<250 ms) and late phase (>250 ms). (e) The peak amplitude was significantly larger in MCs (n = 10) than in TC (n = 10) during both the early and late phases of inhibition. (f) Charge transferred was significantly larger in MCs than TCs during the late phase only. ('x' indicates cells lacking apical dendrites). (g) MCs and TC recorded sequentially in the same slice (n = 5 slices) show similar differences in inhibition. (h–i) Analogous experiment measuring lateral inhibition in OMP-ChR2-YFP mice before and after limiting GC-mediated inhibition by bath applying APV and LY36785. (j,k) Similar differences in the peak amplitude (j) and charge (k) in MCs (n = 5) and TCs (n = 5) were found. (l) Ratio of early and late phase amplitude and charge after and before limiting GC-mediated inhibition. Data are presented as mean ± s.e.m. Statistical tests in e,f,j,k,l were two-tailed, unpaired t tests and tests in g were paired t tests. (*p<0.05, **p<0.01, ***p<0.001)

The following figure supplement is available for figure 1:

**Figure supplement 1.** Distance dependence of lateral inhibition onto MCs and TCs.

of firing observed following in vivo odor delivery (*Patterson et al., 2013*) or in vitro glomerular activation (*Najac et al., 2015*) (*Figure 2—figure supplement 1*). Similar proportions of MCs (16/25–64%) and TCs (12/18–67%) were affected by lateral inhibition (see Materials and methods), and unaffected cells were excluded from further analysis. Additionally, MCs and TCs were similar distances from the M72 glomerulus (*Figure 3—figure supplement 1*). Similar to our previous results using paired MC recordings (*Arevian et al., 2008*), MC firing rates were reduced by lateral inhibition selectively when MCs fired at intermediate rates, although inhibition was observed at more than double the rate observed in paired recordings. For example, the MC in *Figure 2b,e* was affected by lateral inhibition while firing between 27–76 Hz. Likewise, across a population of 16 MCs, rates between 34 ± 11 Hz and 79 ± 22 Hz were affected by lateral inhibition (*Figure 3a–e*, *Figure 3—figure supplement 2a–c*). Surprisingly, TCs were influenced by lateral inhibition only when firing at low rates. The example TC in *Figure 2c,d,f* was affected by lateral inhibition while firing between 5–42 Hz, and across a population of 12 TCs, rates between 10 ± 4 Hz and 43 ± 8 Hz were affected by lateral inhibition (*Figure 3a–e*, *Figure 3—figure supplement 2d–f*). Both the lower and upper bounds of the effective activity range of lateral inhibition (i.e. the range of firing rates over which lateral inhibition reduces firing rates) were significantly lower in TCs than in MCs (*Figure 3e*).

The average fractional reduction in firing rate was much larger in TCs than in MCs (*Figure 3c*), though the absolute firing rate decrease was not significantly different (MC: −8.2 ± 2.5 Hz n = 16, vs TC: −8.8 ± 3.2 Hz n = 12) (*Figure 3d,f*). Similar to the findings reported above, neither the effective activity range of lateral inhibition nor the decrease in absolute firing rate depended on whether the cell had an intact apical dendrite. Additionally, these effects of lateral inhibition on firing rate were present even in short timescales matching a single 4 Hz sniff (*Figure 3—figure supplement 3*). Collectively, our results demonstrate that lateral inhibition is functionally distinct in MCs and TCs.

This difference in activity-dependent lateral inhibition (ADLI) may arise from differences in the lateral inhibitory circuits engaged by MCs and TCs. Our finding that inhibition is largely unaffected by apical dendrite truncation suggests that differences in ADLI between MCs and TCs most likely involves inhibitory circuitry within the external plexiform layer (EPL). Consistent with MCs and TCs engaging distinct lateral inhibitory circuits within the EPL, classical morphological studies suggest that GCs are subdivided into superficial GCs (sGCs), which innervate the superficial EPL, and deep GCs (dGCs), which innervate the deep EPL (*Mori et al., 1983*; *Orona et al., 1983*). This putative morphological subdivision of GCs suggests that sGCs inhibit TC lateral dendrites in the superficial EPL while dGCs inhibit MC lateral dendrites in the deep EPL (*Mori et al., 1983*; *Orona et al., 1984*). Therefore, functional differences between sGCs and dGCs may mechanistically underlie the difference in ADLI observed between MCs and TCs. In particular, given our prior results demonstrating that low firing rates in MCs are unaffected by lateral inhibition because many GCs require cooperative inputs from multiple glomeruli in order to be activated (*Arevian et al., 2008*), we hypothesized that low firing rates in TCs are affected by lateral inhibition because sGCs are more strongly recruited than dGCs following activation of a single glomerulus.

Several previous findings are consistent with this hypothesis. GC soma position correlates with GC subtype (*Mori et al., 1983*; *Orona et al., 1983*) and odor-evoked activity in vivo (*Wellis and Scott, 1990*). Specifically, GCs located in the MCL and upper GCL tend to exhibit sGC

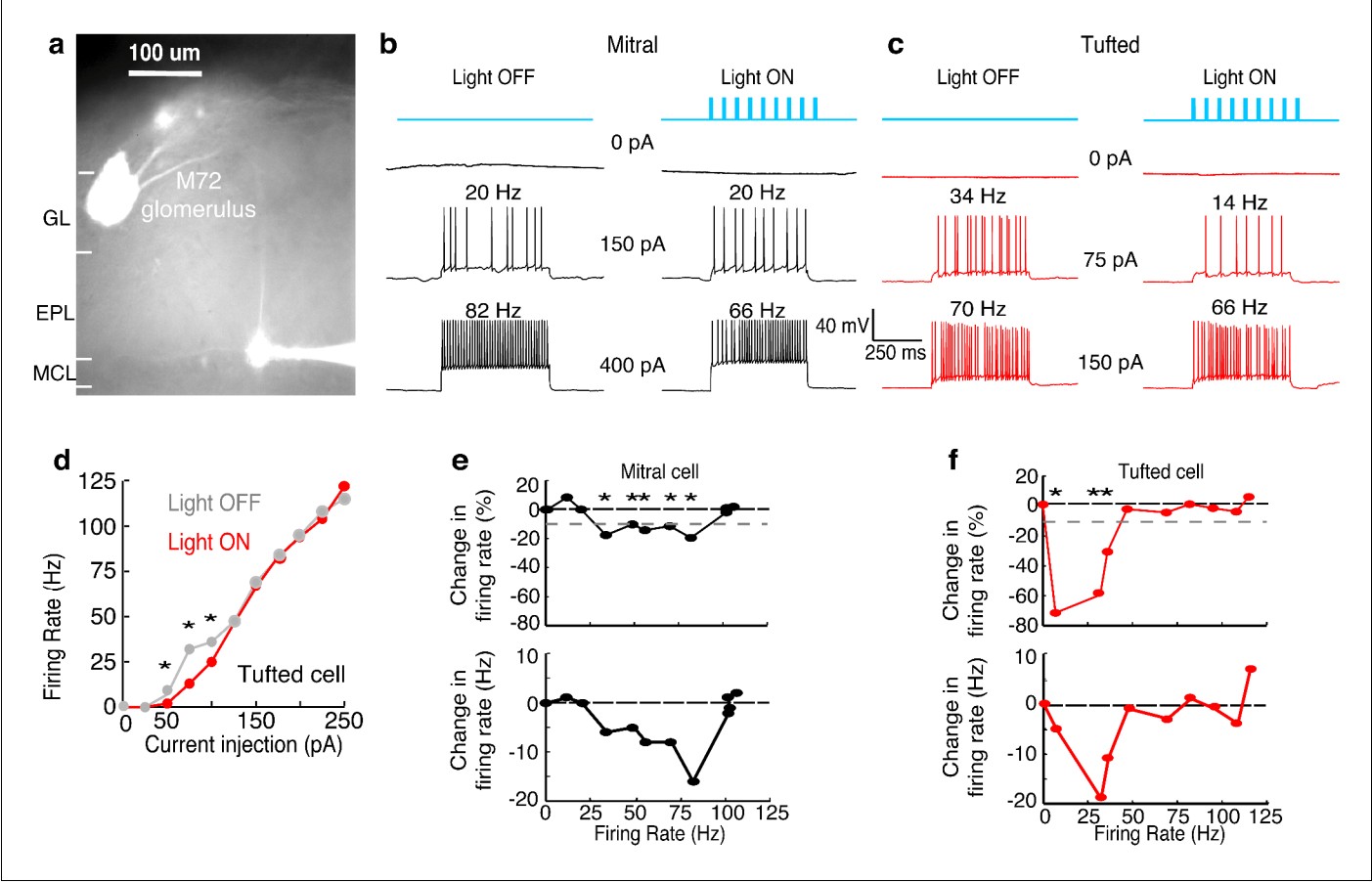

**Figure 2.** Examples show a TC and MC affected by lateral inhibition at low and intermediate firing rates, respectively. (GL: glomerular layer, EPL: external plexiform layer, MCL: mitral cell layer). (a) Fluorescent image of a recorded MC that projects to a glomerulus near the M72 glomerulus. (b–f) The impact of lateral inhibition on one example MC (b, e) and one example TC (c, d, f) was assessed by constructing FI curves for each cell via somatic current injections of increasing amplitudes. At each current step, the number of action potentials evoked with and without M72 photostimulation was determined (10 ms pulses at 15 Hz). (b–c) Examples of voltage traces in a MC (b) or TC (c) at 2 different firing rates with and without M72 photostimulation. (d) The effect of lateral inhibition is illustrated by comparing the FI curves for the light off (grey) versus light on (red) trials in the TC. (e–f) Plots of the percent decrease in firing rate (TOP) or absolute firing rate (BOTTOM) in light on trials for a MC (e) or TC (f). Asterisks signify firing rates that are reduced by more than 10% in at least 2 consecutive light on trials.

The following figure supplement is available for figure 2:

**Figure supplement 1.** MCs and TCs can sustain high firing rates (>50 Hz) for long periods (>500 ms) following glomerular stimulation.

morphologies and suprathreshold odor responses while GCs located in the lower GCL tend to exhibit dGC morphologies and subthreshold odor responses (*Mori et al., 1983*; *Orona et al., 1983*; *Wellis and Scott, 1990*). Whether these differences in odor-evoked activity in vivo reflect functional differences between GC subtypes and M/TC connectivity remains unclear, however, as: 1) odors activate multiple glomeruli in distinct spatiotemporal patterns, which can evoke a complex array of convergent excitation and inhibition onto individual GCs (*Burton and Urban, 2015*), and 2) GC activity in vivo is strongly influenced by centrifugal input (*Balu et al., 2007*; *Boyd et al., 2012*; *Markopoulos et al., 2012*) and anesthesia (*Kato et al., 2012*; *Cazakoff et al., 2014*).

Therefore, to more directly test our above hypothesis, we recorded the excitatory synaptic input and spiking response of GCs to activation of single nearby glomeruli, with post-hoc recovery of Neurobiotin-filled cell morphologies. For this experiment, we switched from selective optogenetic stimulation of M72 glomeruli to extracellular stimulation of untagged glomeruli in order to test a large number of GC-glomerulus pairs. While this approach introduces a degree of glomerulus-to-

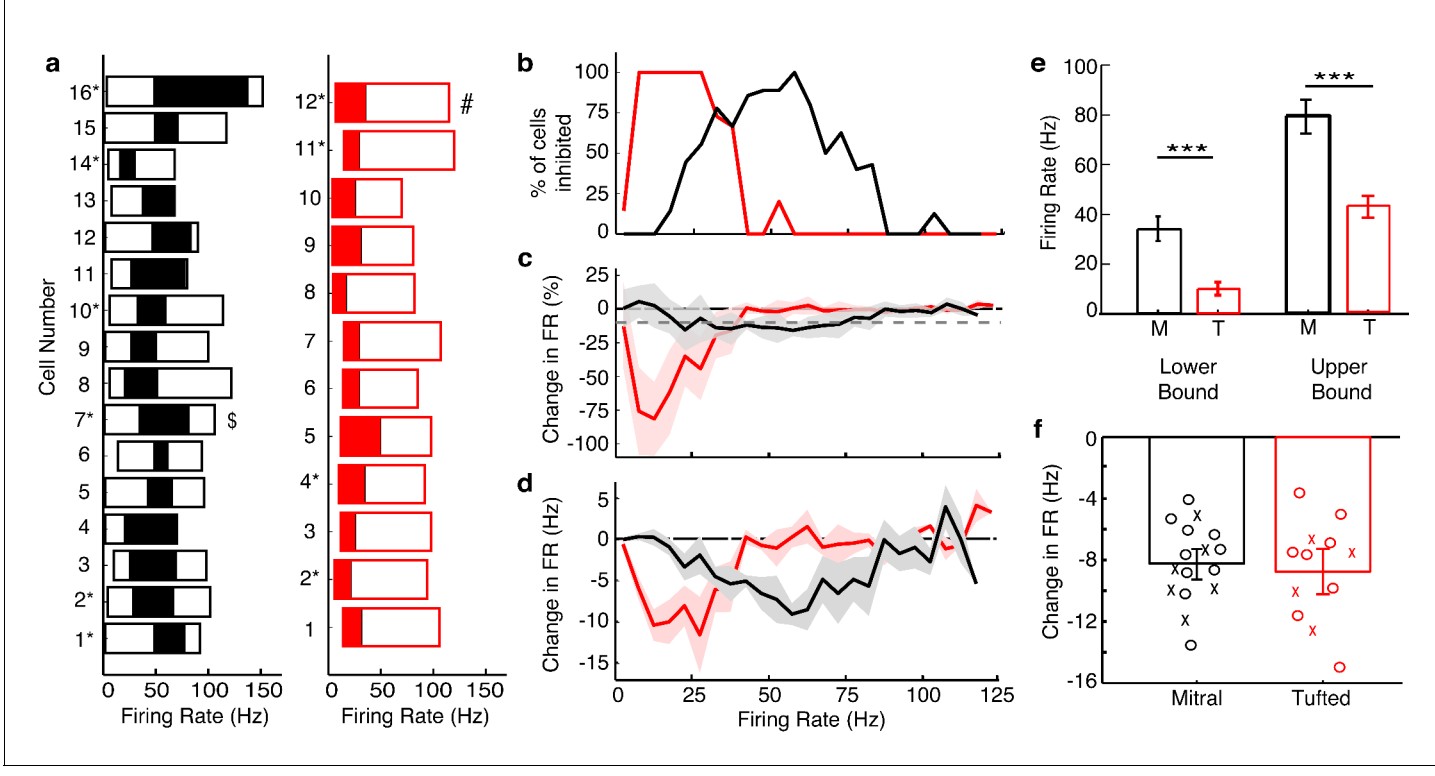

**Figure 3.** Summary results show that TCs are influenced by lateral inhibition at low rates while MCs are influenced at intermediate rates. (a) The range of firing rates that are significantly influenced by lateral inhibition is plotted with respect to the firing rate in light off trials. Outer rectangle for each cell indicates the total range of firing rates evoked during light off trials. Inner (shaded) rectangle indicates the range of firing rates that are reduced during light on trials. Asterisks indicate cells that lack apical dendrites. ($ - MC used in **Figure 2**, # - TC used in **Figure 2**). (c) Percentage of cells that are significantly inhibited is plotted with respect to the firing rate in light off trials. (c–d) Average decrease in firing rate (c – plotted as percent, d – plotted as Hz) across the population of MCs (black) and TCs (red). Grey dotted line in c represents the 10% threshold used to indicate significant inhibition. Shaded areas represent s.d. (e) The lower (LEFT, unpaired t-test, p = 2.8 × 10$^{-7}$) and upper bound (RIGHT, unpaired t-test, p = 1.7 × 10$^{-5}$) of the range of rates affected by lateral inhibition are significantly lower in TCs compared to MCs. (f) There is no change in the average decrease in firing rate (Hz) between MCs and TCs (unpaired t test, p = 0.87). In f, decreases in firing rate were calculated as the average decrease across all significantly affected firing rates. Data are presented as mean ± s.e.m.

The following figure supplements are available for figure 3:

**Figure supplement 1.** Distance dependence of lateral inhibition onto MCs and TCs.

**Figure supplement 2.** Examples of FI curves with and without photostimulation of the M72.

**Figure supplement 3.** The effects of lateral inhibition are maintained on the physiologically relevant timescale of a single 4 Hz (250 ms) sniff.

glomerulus variability to our data, we capitalized on the well-established all-or-none nature of glomerular activation at low stimulation intensities (*Carlson et al., 2000*; *Gire and Schoppa, 2009*) to enable across-cell comparisons of synaptic input and spiking responses following glomerular activation. In addition, we and others have previously demonstrated that optogenetic photostimulation and extracellular stimulation of OSN axons trigger comparable sensory-evoked input to both M/TCs and GCs (*Gire et al., 2012*; *Burton and Urban, 2014*, *2015*).

Consistent with previous morphological accounts (*Mori et al., 1983*; *Orona et al., 1983*), GCs exhibited distinct sGC or dGC morphologies upon visual inspection (*Figure 4a,d*; *Figure 4—figure supplement 1*). Indeed, reconstruction and visual classification of a large subset of recorded GCs and analysis of the spatial distribution of gemmules – the site of reciprocal dendrodendritic synapse formation (*Rall et al., 1966*) – confirmed that sGCs preferentially innervate the superficial EPL while dGCs preferentially innervate the deep EPL (*Figure 4g*). Moreover, somatic depth significantly – but

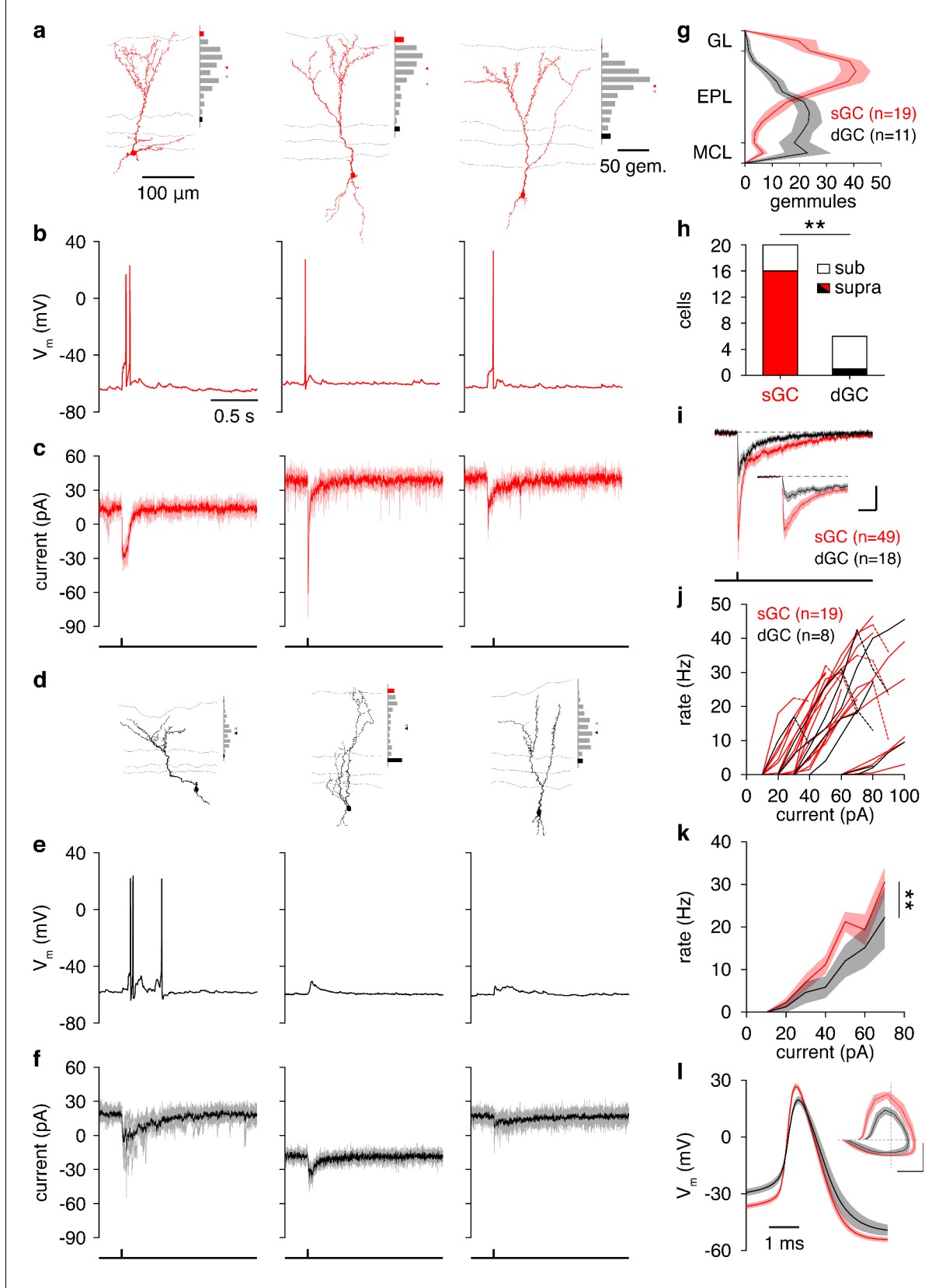

**Figure 4.** Synaptic and intrinsic differences regulate sGC vs. dGC recruitment following glomerular activation. (a) Reconstructed morphologies and distribution of apical dendritic gemmules across the MCL (black bars), EPL (grey bars), and GL (red bars) of 3 representative sGCs. Grey/red ticks

*Figure 4 continued on next page*

*Figure 4 continued*

represents the midpoint of the EPL/mean of the cell's gemmule distribution. (**b,c**) Spiking response (**b**) and synaptic input (**c**) of the 3 sGCs shown in **a** following activation of a single glomerulus superficial to the targeted GC. (**d**−**f**) Same as a-c for 3 representative dGCs. (**g**) Distribution of apical dendritic gemmules across reconstructed sGCs and dGCs. (**h**) A greater proportion of sGCs than dGCs fired in response to glomerular activation (Chi-square test, p = $4.2 \times 10^{-3}$). (**i**) Excitatory input to sGCs exhibited larger peak currents (rank-sum test, p = $8.5 \times 10^{-3}$) and charge transferred (rank-sum test, p = 0.046) than excitatory input to dGCs. No difference in excitation latency was observed (6.6 ± 11.6 vs. 9.7 ± 11.4 ms; rank-sum test, p = 0.12). Scalebar: 0.2 s/10 pA (inset: 40 ms/20 pA). (**j,k**) sGCs and dGCs showed significantly different firing rate-current (FI) curves in response to somatic step current injection (2-way ANOVA, p = $4.1 \times 10^{-3}$). Individual (**j**) and mean (**k**) FI curves shown. Dashed lines show diminished firing due to depolarization block. (**l**) sGC action potentials exhibited more hyperpolarized thresholds (unpaired *t* test, p = $4.8 \times 10^{-3}$), larger amplitudes (unpaired *t* test, p = $1.1 \times 10^{-4}$), and faster rising slopes (unpaired *t* test, p = $2.9 \times 10^{-4}$) than dGC action potentials. Inset: action potential phase plot. Scalebar: 30mV/ 100 mVms$^{-1}$; dashed lines show origin. Shaded regions show mean ± SEM.

The following source data and figure supplements are available for figure 4:

**Source data 1.** Tables of GC properties.
**Figure supplement 1.** Morphological analysis of superficial and deep GCs.
**Figure supplement 2.** Proposed mechanism of activity-dependent lateral inhibition in mitral (MCs) and tufted cells (TCs).

incompletely – predicted GC subtype (*Figure 4a,d*; *Figure 4—figure supplement 1*; *Figure 4—source data 1*, Table 1), as previously observed (*Mori et al., 1983*; *Orona et al., 1983*). To determine whether the observed morphological differences reflect subtypes of GCs rather than a continuum, we additionally performed unbiased clustering of GCs. Specifically, clustering of GCs by the Euclidean distances among their normalized gemmule distributions (using Ward's method) and application of the gap statistic method yielded 3 distinct clusters: dGCs, sGCs, and a small group of sGCs with prominent innervation of the deep glomerular layer (*Figure 4—figure supplement 1c*). Moreover, these clusters closely aligned with our original classification by visual inspection, with 19 of 19 sGCs and 9 of 11 dGCs correctly assigned (*Figure 4—figure supplement 1c*). Our results therefore quantitatively confirm the morphological subdivision of GCs into distinct subclasses of sGCs and dGCs.

In agreement with our hypothesis, a strikingly higher percentage of sGCs than dGCs fired in response to activation of a single glomerulus (*Figure 4b,e,h*) due, at least partially, to stronger excitatory synaptic input to sGCs than dGCs (*Figure 4i*). As a caveat, we note that the greater recruitment of sGCs following glomerular activation may arise as an artifact of our acute slice preparation. Specifically, as TC circuitry is closer to any given glomerulus than MC circuitry, TC-mediated input to GCs (likely sGCs) may be better preserved than MC-mediated input to GCs (likely dGCs) in the acute slice, leading to stronger sGC excitation and recruitment following glomerular activation. However, despite the fact that these observations were made in vitro, three lines of evidence support greater feedforward recruitment of sGCs as a physiological feature of the olfactory bulb circuit. First, our in vitro observation of greater sGC firing following glomerular activation (*Figure 4b,e,h*) corresponds well with the previous in vivo observation of stronger odor-evoked activity in putative sGCs (*Wellis and Scott, 1990*). Second, examination of GC biophysical properties revealed several intrinsic differences supporting greater recruitment of sGCs than dGCs, including a more hyperpolarized action potential threshold in sGCs (*Figure 4l*; *Figure 4—source data 1*, Table 2) and greater intrinsic excitability in sGCs in response to somatic step current injections (*Figure 4j,k*; *Figure 4—source data 1*, Table 3), despite equivalent somatodendritic sizes (*Figure 4—figure supplement 1d*; *Figure 4—source data 1*, Table 1) and passive membrane properties (*Figure 4—source data 1*, Table 4) between sGCs and dGCs. Third, analysis of spontaneous synaptic activity revealed no difference in event frequency or amplitude between sGCs and dGCs (*Figure 4—source data 1* Table 5). Critically, recordings of spontaneous synaptic activity were performed in the absence of TTX and thus contain some degree of action potential-dependent input, which likely originates from intact presynaptic cells. Therefore, equal spontaneous event frequencies between sGCs and dGCs suggests that their respective presynaptic circuits are comparably intact. Moreover, equal spontaneous event amplitudes suggest a comparable contribution of larger action potential-dependent and smaller action

potential-independent events between sGCs and dGCs, again consistent with comparably intact presynaptic circuits. In total, our results thus support the hypothesis sGCs are more strongly recruited than dGCs following activation of a single glomerulus due to stronger excitatory input and greater intrinsic excitability, although we cannot rule out an important role for other mechanisms, especially in the intact system in vivo.

As an additional test of our hypothesis that differences in the excitability of GCs provide one potential mechanism underlying the functional difference in ADLI between MCs and TCs, we measured the effects of lateral inhibition on MC spiking before and after increasing the excitability of GCs with the mGluR agonist (RS)-3,5-dihydroxyphenylglycine (DHPG, 10 µM) (*Dong et al., 2007*; *Heinbockel et al., 2007*). At this concentration, DHPG selectively enhances the excitability of GCs (*Heinbockel et al., 2004*) but not MCs (*Figure 5—figure supplement 1*). The effective activity range of lateral inhibition in MCs (n=6) fell from 31 (± 9)–73 (± 10) Hz to 16 (± 12)–49 (± 23) Hz after adding DHPG (*Figure 5a–d*), and the lower and upper bounds of the effective activity range of lateral inhibition were significantly reduced after the addition of DHPG (*Figure 5e*). Additionally, these effects did not depend on whether apical dendrites were intact (*Figure 5d*), and therefore does not reflect effects of DHPG on glomerular layer circuitry. Despite these observations, however, we note

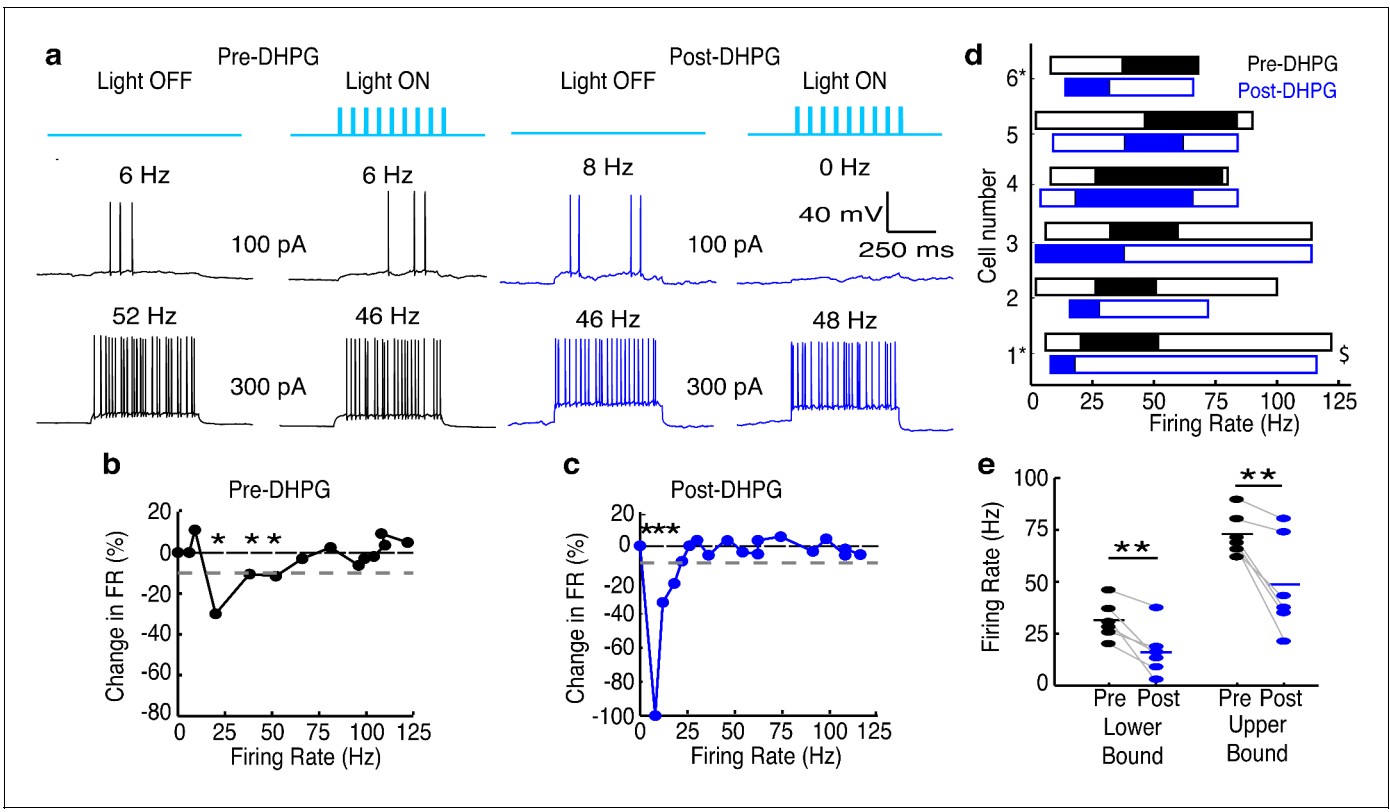

**Figure 5.** Increasing GC excitability shifts the effective activity range of lateral inhibition in MCs to lower frequencies. (a–c) Example of lateral inhibition in one example MC before and after bath applying mGluR agonist, DHPG (10 µM). (a) Example voltage traces from one MC. LEFT: Before application of DHPG, intermediate firing rates are affected by lateral inhibition. RIGHT: After application of DHPG, low firing rates are affected. (b–c) The effect of lateral inhibition is illustrated by comparing the FI curves for the light off versus light on trials in a MC before (b) and after (c) applying DHPG. Plot of the percent decrease in firing rate vs. the firing rate of light off trials. Dotted line in b,c represents the 10% threshold used to indicate significant inhibition. Asterisks signify firing rates which are reduced by more than 10% in light on trials. (d) Summary of 6 cells recorded before (black) and after (blue) bath application of DHPG ($ - cell depicted in a–c). (e) The lower (LEFT, paired t-test, p = 0.009) and upper bound (RIGHT, paired t-test, p = 0.009) of the effective activity range of lateral inhibition are significantly lower after application of DHPG.

The following figure supplement is available for figure 5:

**Figure supplement 1.** DHPG has no effect on MC excitability.

that increased GC excitability may not be the sole mechanism responsible for these DHPG-induced shifts in the effective range of lateral inhibition. For instance, DHPG-induced increases in the excitability of external tufted cells (*Dong and Ennis, 2014*) may, in turn, increase the activity in M72-MCs, the activity in GCs and GC-mediated inhibition onto the recorded MC. These data (*Figures 4–5*) provide two indirect lines of evidence that suggest that differences in GC populations provide one mechanism for differences in the effective activity range of lateral inhibition onto MCs and TCs.

Together these data suggest that distinct odor-evoked activity observed in MCs and TCs in vivo arises, in part, due to differences in how ADLI affects each cell type (*Figure 4—figure supplement 2*). To begin assessing how these circuit-level differences between MCs and TCs impact their ability to encode olfactory information, we performed simulations of MC and TC networks to determine how differences in ADLI may translate into differences in stimulus encoding.

We simulated an olfactory discrimination task in which a presented odor must be identified from a panel of similar odors. Because the goal of the simulations was to understand how the differences in MC and TC circuit properties described above influence population coding, we used simple firing rate models in which we could directly and independently modify lateral inhibition properties without changing other model features. We therefore made relatively few assumptions in performing our simulations, mostly relating to how odor concentration is encoded (as firing rate differences [*Cang and Isaacson, 2003*; *Fukunaga et al., 2012*; *Igarashi et al., 2012*; *Sirotin et al., 2015*] – however see [*Meredith, 1986*]), how lateral inhibitory connectivity is specified (randomly), and how MCs and TCs differ (which was explicitly explored). Each odor was simulated as a pattern of inputs to populations of MCs or TCs, corresponding to activated glomeruli (*Figure 6a,b*, *Figure 6—figure supplement 1*). Differences in odor concentration were modeled as changes in the number and intensity of activated glomeruli (*Rubin and Katz, 1999*; *Meister and Bonhoeffer, 2001*) (see Materials and methods). Additionally, we added trial-to-trial variability in each presentation of a particular odor (*Figure 6b*, see Materials and methods) in order to mimic natural fluctuations in the background odors and in the patterns of odor-evoked glomerular activation (*Wachowiak et al., 2004*).

In our simulations, we asked how well MC and TC population activity discriminated between similar odors presented at the same odor concentration. While understanding how animals discriminate between odors presented at a variety of concentrations is important, we confined our discriminations to odors presented at the same concentration to more closely match behavioral experiments in mice (*Abraham et al., 2010*; *Lepousez and Lledo, 2013*) and to keep from making a number of assumptions about how the representation of individual odors varies with concentration. In this simulation, each MC or TC received excitatory input from one of the 150 glomeruli. On each trial, we randomly presented one odor from the panel to networks comprised entirely of either MCs or TCs. We applied a common decoding algorithm, linear discriminant analysis, to determine the extent to which different odors were discriminable in our simulated MC and TC populations (*Figure 6c*). While the strategy that downstream brain areas use to decode information contained in MC and TC outputs is unknown, linear discriminant analysis is a simple classification algorithm, has some degree of biological plausibility and has been widely applied in similar contexts (*Quiroga et al., 2007*; *Quian Quiroga and Panzeri, 2009*; *Giridhar et al., 2011*).

We first compared the discrimination accuracy of three models: one in which ADLI affected low rates (i.e. an all TC network), one in which ADLI affected intermediate rates (i.e. an all MC network), and a control population that lacked any inhibition. ADLI was modeled by explicitly adjusting the sensitivity of MC and TC firing rates to inhibition without changing other parameters of the model (*Figure 6d–e*). When the odor panel consisted of only 8 odors, both MC and TC populations discriminated between odors with an accuracy that did not differ from the control population that lacked inhibition (*Figure 6f*). However, when we increased the difficulty of the task by increasing the size of the odor panel (to 32 odors), the difference in accuracy between MCs and TCs dramatically increased (*Figure 6g*). For the set of 32 odors, TCs significantly outperformed MCs and the control population at low odor concentrations (90% accuracy in TCs compared to 70% accuracy in MCs at 30% of maximum concentration) while MCs significantly outperformed TCs and the control population at high concentrations (73% accuracy in TCs compared to 91% accuracy in MCs at 60% of maximum concentration). Therefore, differences in ADLI alone support concentration-dependent differences in odor discrimination.

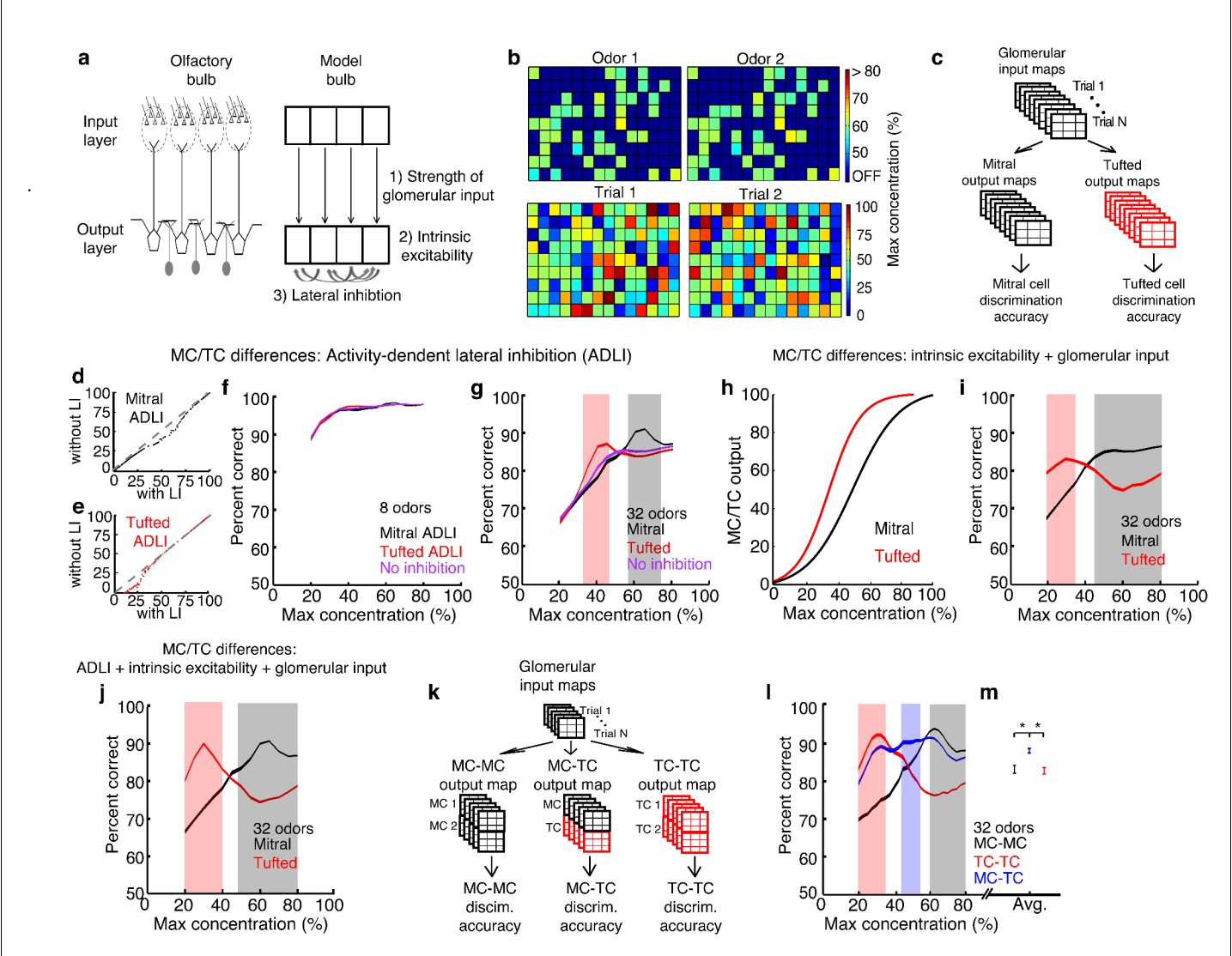

**Figure 6.** MCs and TCs discriminate between odors best in separate concentration ranges. (a) Odors are defined as the pattern of activated glomeruli (i.e. pixels) in the model. Each glomerulus provides input to only 1 MC/TC. (b) TOP: Pattern of activated glomeruli for 2 odors. BOTTOM: Two separate presentations of Odor 1. (c) Outputs of MCs or TCs are used to train and test a linear classifier to predict which odors are presented on each trial. (d– e) Lateral inhibition differences between MCs and TCs are modeled using Gaussian distributions centered at different firing rates. Firing rates of MCs (d) and TCs (e) with and without lateral inhibition. (f–g) Discrimination accuracy of 8 (f) and 32 (g) odors in MCs (black), TCs (red) and a control population of neurons that lacked any inhibition (purple). Shaded areas represent concentration ranges where TCs (light red) or MCs (grey) discriminate significantly better (see Online methods). (h) Differences in the strength of excitatory inputs and intrinsic excitability between MCs and TCs are modeled using 2 sigmoids to translate glomerular inputs into MC/TC outputs. (i) Discrimination accuracy of 32 odors in MCs (black) and TCs (red) that differ in excitability, strength of glomerular input. (j) Discrimination accuracy of 32 odors in MCs (black) and TCs (red) that differ in excitability, strength of glomerular input and ADLI. (k) Overview of simulations comparing 3 separate output neuron configurations: 2 MCs per glomerulus, 2 TCs per glomerulus or 1 MC and 1TC per glomerulus. (l) Discrimination accuracy of 32 odors for MC-MC (black), TC-TC (red) or MC-TC (blue) networks across a range of concentrations. (m) Average discrimination accuracy across all concentrations plotted in l (*p<1e-4). Width of plots in panels f,g,i,j,l reflect the s.e.m.

The following figure supplements are available for figure 6:

**Figure supplement 1.** Procedure used to create the odors that served as inputs to simulated mitral and tufted cell networks.

**Figure supplement 2.** Subtractive or divisive lateral inhibition does not improve discrimination accuracy.

*Figure 6 continued*

**Figure supplement 3.** Visual processing example of how the multiple parallel neuron populations can simultaneously enhance the contrast of high and low intensity images.

Next we tested how well different forms of activity-independent inhibition compare to ADLI in their ability to improve discrimination accuracy. Specifically, we compared discrimination accuracy (in panels consisting of 32 odors) between 3 populations of neurons; one with subtractive inhibition (*Figure 6—figure supplement 2a*), one with divisive inhibition (*Figure 6—figure supplement 2b*) and a control population that lacked inhibition. We found that populations using subtractive or divisive inhibition performed no better than the control population across all concentrations (*Figure 6—figure supplement 2c*). These results agree with prior work that showed that ADLI substantially decorrelates MC responses to similar odors while subtractive and divisive forms of lateral inhibition have little effect on MC correlation (*Arevian et al., 2008*).

Prior work has shown that, in addition to ADLI differences, TCs are more intrinsically excitable (*Burton and Urban, 2014*) and receive stronger OSN inputs than MCs (*Gire et al., 2012*; *Burton and Urban, 2014*). Therefore, we next asked how well these differences between MCs and TCs affect odor discrimination. Differences in the strength of OSN input and intrinsic excitability were modeled by altering sigmoid functions that relate the intensity of glomerular inputs to M/TC outputs (*Figure 6h*). These differences in MC vs. TC excitability and OSN input strength also allow TCs and MCs to discriminate odors best at low and high concentrations respectively (*Figure 6i*). Finally we simulated how all three differences between MCs and TCs – ADLI, intrinsic excitability and glomerular input – affected odor concentration. Intriguingly, TCs were best at discriminating between low concentration odors when all three differences were included in the model (*Figure 6j*). Together, these simulations show that intrinsic and circuit-level differences between MCs and TCs work cooperatively to optimize discrimination between similar odors in separate concentration ranges.

Finally, we asked more generally how sensory systems, including the olfactory system, might benefit from splitting information into multiple channels with distinct ADLI. First, we asked whether models containing a combination of MCs and TCs discriminate odors better than models that have only MCs or TCs. We used a variant of the models presented above: here, each glomerulus provided input to 2 neurons instead of 1. We compared three different output neuron configurations: 2 MCs per glomerulus, 2 TCs per glomerulus, and 1 MC and 1 TC per glomerulus (*Figure 6k*, see Materials and methods). Similar to the data presented in *Figure 6g*, TC models discriminated between 32 odors best at low concentrations and MC models were best at high concentrations (*Figure 6l*). However, MC-TC models significantly outperformed other models at intervening odor concentrations. Interestingly, MC-TC models also exhibited the second best performance at high and low concentrations, and after averaging discrimination accuracy across all concentrations tested, we found that models containing a combination of MCs and TCs performed significantly better than models composed of only MCs or only TCs (*Figure 6m*). Second, new features might arise in any brain area that implements ADLI and splits information into parallel channels. Using an image processing analogy, we show that if an image is split into parallel channels and then recombined by a downstream population, new information about the image (i.e. higher contrast), absent in either single channel, emerges (*Figure 6—figure supplement 3*). Therefore systems that split sensory information into multiple pathways can not only outperform single pathway systems at comparable tasks, but are also capable of performing novel computations unlikely to occur in single pathway systems.

## Discussion

Parallel pathways are a common feature of many sensory systems. Yet how local circuit activity creates stimulus selectivity in parallel pathways remains poorly understood. Here, we have identified a novel circuit mechanism for generating differential responses across two parallel pathways (formed by MCs and TCs) in the olfactory system and examined the emergence of feature selectivity. We find that differences in ADLI selectively reduce intermediate firing rates in MCs and low firing rates in TCs. We provide evidence that these effects could be caused, in part, by differences in the

excitability of subclasses of GCs that preferentially inhibit TCs vs. MCs. Moreover, using simulations, we show that differences in the effective activity range of lateral inhibition, along with other intrinsic and circuit-level differences between MCs and TCs, work cooperatively to enable MCs and TCs to best discriminate between similar odors in separate concentration ranges. Finally, we show that the combination of MCs and TCs facilitates odor discrimination across a wide range of odor concentrations.

The activity-dependence of lateral inhibition depends on three cell populations – the 'presynaptic' M/TCs associated with the M72 glomerulus (which we activate optogenetically via stimulation of OSNs), the inhibitory interneurons and the 'postsynaptic' M/TCs. Our prior work has shown that increasing the firing rate of presynaptic MCs (the M72 MCs in this case) shifts the range of MC firing rates influenced by lateral inhibition (*Arevian et al., 2008*). Given that TCs fire at higher rates than MCs following glomerular activation (*Burton and Urban, 2014*), M72 TCs will fire at higher rates than M72 MCs following M72 photoactivation and, in part, contribute to the decreased lower bound of the effective activity range of lateral inhibition in postsynaptic TCs. However, higher firing rates in M72 TCs alone is insufficient to explain the differences in both the lower and upper bound in the effective activity range of lateral inhibition between MCs and TCs.

Our analysis supports the hypothesis that TCs and MCs are preferentially connected to sGCs and dGCs, respectively. Additionally, we provide indirect evidence that differences in the recruitability of sGCs and dGCs may be one potential mechanism underlying the differences in ADLI between MCs and TCs. Our data suggests that sGCs spike with a higher probability than dGCs following the activation of a single glomerulus. Consequently, postsynaptic TCs can effectively recruit lateral inhibition when firing at low rates. dGCs are less excitable and therefore require more input to be activated. Therefore M72 photostimulation alone causes relatively weak activation of dGCs. However, when the postsynaptic MC fires at intermediate rates, additional dGCs will become activated and mediate lateral inhibition. In support of this mechanism, we show that increasing the excitability of GCs by activating mGluRs (*Dong et al., 2007*; *Heinbockel et al., 2007*) shifts the effective activity range of lateral inhibition in MCs to lower firing rates (*Figure 5*). Additionally, these differences between sGCs and dGCs can explain why the upper bound on the effective activity range of lateral inhibition differs between MCs and TCs. Cells firing above this upper bound recruit the maximum amount of recurrent inhibition such that additional inputs to GCs cannot trigger additional GABA release (*Urban and Arevian, 2009*). Because sGCs are more excitable than dGCs, maximal GABA release, and consequently the upper bound on the effective activity range of lateral inhibition, occurs at lower rates in TCs than in MCs (*Figure 4—figure supplement 2*).

While preferential connectivity of TCs with sGCs and MCs with dGCs can explain the observed differences in the activity-dependence of lateral inhibition, other circuit mechanisms may also contribute. In particular, EPL interneurons (EPL-INs) can mediate inhibition onto M/TCs (*Huang et al., 2013*; *Kato et al., 2013*; *Miyamichi et al., 2013*), and it remains possible that EPL-INs or distinct subclasses of EPL-INs preferentially inhibit MCs or TCs. However, it is unlikely that EPL-INs predominantly drive the difference in MC vs. TC ADLI given that EPL-INs mediate linear (divisive) but not activity-dependent inhibition of MC/TC outputs (*Kato et al., 2013*; *Uchida et al., 2013*). Glomerular layer circuits have also been shown to play a role in M/TC lateral inhibition (*Aungst et al., 2003*; *Liu et al., 2013*; *Whitesell et al., 2013*; *Banerjee et al., 2015*), however these circuits likely do not play a role in the differences in ADLI reported here because our results were not influenced by apical dendrite truncation. Therefore, distinct GC populations that differ in excitability are the most parsimonious explanation for the observed differences in ADLI.

Differential connectivity of MCs and TCs with distinct GC subclasses has long been predicted based on the putative morphological subdivisions of GC apical dendritic morphologies (*Mori et al., 1983*; *Orona et al., 1983*) and separation of MC and TC lateral dendrites in the deep and superficial EPL (*Mori et al., 1983*; *Orona et al., 1984*), respectively. However, distributed M/TC-GC connectivity along exceedingly long M/TC lateral dendrites has, thus far, precluded a direct demonstration of differential connectivity of MCs and TCs with distinct GC subclasses. Here, we provide quantitative evidence that GCs indeed form two distinct morphological subclasses – sGCs and dGCs – rather than a continuum of morphologies. Moreover, our results provide – to our knowledge – the first functional evidence that MCs and TCs engage distinct lateral inhibitory circuits, likely via differential connectivity with dGCs and sGCs, respectively. Future experiments involving selective manipulation

of MCs vs. TCs and dGCs vs. sGCs will ultimately be needed in order to directly demonstrate whether (and the degree to which) these lateral inhibitory circuits overlap.

Our data describe local, circuit-level mechanisms that can account for several differences between MC and TC odor-evoked responses observed in vivo. Weaker lateral inhibitory currents may contribute to the finding that TCs are less frequently inhibited by odors (*Nagayama et al., 2004*). Additionally, the finding that intermediate and high firing rates are unaffected by lateral inhibition in TCs may help explain why TCs show odor-evoked responses that are more highly correlated to OSN input (*Adam et al., 2014*) and less dependent on concentration (*Fukunaga et al., 2012*; *Igarashi et al., 2012*). Additionally, TCs firing above the upper bound of the effective activity range of lateral inhibition are shielded, not only from lateral inhibition, but other sources of GC-mediated inhibition as well, such as inhibition triggered by cortical feedback (*Boyd et al., 2012*; *Markopoulos et al., 2012*), Consequently, the circuit-level differences we describe may explain why MCs but not TCs are decorrelated by cortical feedback (*Otazu et al., 2015*).

These circuit-level differences between MCs and TCs likely affect olfactory discrimination (*Figure 6*). ADLI, but not subtractive or divisive inhibition, allows TCs and MCs to best discriminate between odors at low and high concentrations, respectively. Yet how does this improvement in discrimination occur? Prior work has shown that activity in M/TC populations becomes decorrelated over time (*Bathellier et al., 2008*; *Cury and Uchida, 2010*) and that this decorrelation is driven by GCs (*Gschwend et al., 2015*). Additionally, GC-mediated M/TC decorrelation improves odor discrimination (*Gschwend et al., 2015*). Consequently, ADLI likely improves odor discrimination by driving decorrelation of M/TC firing in ranges that engage lateral inhibition (*Arevian et al., 2008*). M/TCs firing at rates outside the effective range of lateral inhibition may be encoding complementary olfactory information beyond odor identity.

Performing these experiments in acute slices provides the best opportunity to explore the causes and consequences of differences in lateral inhibition between MCs and TCs. Given the novel features and mechanisms of ADLI, using a reduced and carefully controlled system in which a single glomerulus can be reliably activated is an important first step. Moreover, our in vitro approach allowed us to apply pharmacological manipulations that were vital in identifying differences in GC excitability (*Figure 5*) rather than differences in glomerular layer circuitry (*Figure 1I*) as the main mechanism supporting differences in MC vs. TC ADLI.

We note, however, that our in vitro approach also has certain limitations. In particular, the slicing procedure may introduce artifacts that could confound our conclusions. Importantly, however, our findings that MCs receive stronger lateral inhibitory currents and are affected at different ranges of firing rates than TCs are unlikely to reflect slicing artifacts. Due to slicing, some MCs and TCs that project to the M72 glomerulus will be truncated, and cells that reside farther from the M72 glomerulus have a higher probability of being truncated. However, our findings do not depend on the cell's distance from the M72 glomerulus (*Figure 1—figure supplement 1*; *Figure 3—figure supplement 1*). Additionally, the lateral dendrites of MCs and TCs innervate distinct strata of the EPL and these differences could result in differences in the fraction of original connections left intact following slicing. However, the effect of ADLI on MCs that are observed here following activation of an entire glomerulus closely mirrors the effects that were observed when activating a single presynaptic MC in our previous work (*Arevian et al., 2008*). That is, only the probability of finding connections and the absolute magnitude of the effect are increased by activating all of the cells associated with a glomerulus (compared to the prior study when only a single presynaptic cell was activated). This suggests that the critical feature of the effect (maximal lateral inhibition at intermediate firing rates) is a property of single connections rather than one that depends on having a full repertoire of connections intact. Moreover, the close correspondence of our GC data to previous in vivo recordings of odor-evoked GC activity (*Wellis and Scott, 1990*), as well as the equivalent spontaneous synaptic activity – a proxy for circuit intactness – observed between sGCs and dGCs suggests that slicing artifacts cannot explain the differences in excitatory input or intrinsic excitability observed between sGCs and dGCs.

Our in vitro approach additionally cannot address multiple important factors that are only present in the intact animal. For instance, respiration and centrifugal feedback likely modulate ADLI in vivo. Addressing the extent to which ADLI affects odor-evoked responses in MCs and TCs in vivo is thus an important future direction.

Finally, what are the benefits of parallel processing? In other sensory systems, different pathways may arise from functional differences at the initial stages of stimulus detection, which are then maintained through selective feedforward connectivity. In the olfactory system there is no evidence that MCs and TCs are targeted by distinct subsets of OSNs. Rather, we show that differences in bulbar circuitry are sufficient to generate important differences in response properties. Here we show that MCs and TCs perform odor discriminations best at separate concentration ranges and that a combination of MCs and TCs discriminates similar odors better than either population alone. These results suggests that parallel processing may offer similar benefits in other sensory systems in which stimulus intensities can vary over many orders of magnitude. For instance, rods and cones in the retina function best at different ranges of light intensity. Other similarities between the olfactory and visual systems suggest that the mechanisms behind parallel processing may be shared across sensory systems. In the retina, feature selectivity in each ganglion cell type emerges due to its connectivity to specific bipolar and amacrine cell types (*Masland, 2012*). Similarly, differences in odor-evoked responses in MCs and TCs emerge due to differences in connectivity to OSNs, external tufted cells (*Najac et al., 2011*; *Gire et al., 2012*) and GCs. Therefore, in these systems, feature selectivity is not inherited but emerges via differential connectivity with distinct neuron types. Similarly, recombining the specific features encoded in individual channels allows new features to emerge. For instance, if populations of neurons in higher order sensory areas integrate inputs from parallel sensory channels, such as the anterior piriform cortex (*Nagayama et al., 2010*; *Igarashi et al., 2012*), and perform simple computations, such as averaging, new features may emerge (*Figure 6—figure supplement 3*). More complex integration mechanisms may allow the calculation of other complex stimulus features using the different information encoded in multiple parallel sensory channels.

## Materials and methods

### Ethical approval

All experiments were completed in compliance with the guidelines established by the Institutional Animal Care and Use Committee of Carnegie Mellon University (IACUC # AS15-010) and University of Pittsburgh (IACUC # 15116582).

### Slice preparation

For MC and TC recordings, postnatal day 16-23 M72-ChR2-YFP (*Smear et al., 2013*) mice were anaesthetized with isoflurane and decapitated into ice-cold oxygenated dissection solution containing (in mM): 125 NaCl, 25 glucose, 2.5 KCl, 25 NaHCO$_3$, 1.25 NaH$_2$PO$_4$, 7 MgCl$_2$ and 0.5 CaCl$_2$. Sagittal slices (280 μm thick) of the MOB were prepared using a vibratome (VT1200S; Leica, Nussloch, Germany). Slices recovered for 15–30 min in 37°C oxygenated Ringer solution that was identical to the dissection solution except for lower Mg$^{2+}$ concentrations (1 mM MgCl$_2$) and higher Ca$^{2+}$ concentrations (2 mM CaCl$_2$). Slices were then stored in room temperature oxygenated Ringer solution until recording. For MC and TC recordings in OMP-ChR2-YFP mice, horizontal slices (280 μm thick) were prepared from postnatal day 17–21 mice. For GC recordings, equivalent methods were used to prepare horizontal slices (310 μm) of the MOB from postnatal day 18–28 C57BL/6, Thy1-YFP-G (*Feng et al., 2000*) albino C57BL/6J, and heterozygous OMP-ChR2-YFP (*Smear et al., 2011*) mice using a vibratome (5000mz-2, Campden).

### Cell identification and morphological analyses

TCs were identified as those cells with large somas (>10 μm in diameter) that reside completely in the EPL. Cell bodies resided in the superficial half of the EPL. All TCs included in our final dataset had at least 1 lateral dendrite and did not display the rhythmic bursting characteristic of external tufted cells (*Hayar et al., 2004*; *Antal et al., 2006*; *Liu and Shipley, 2008*). MCs were identified as large cells located in the mitral cells layer (MCL). 'Displaced MCs (*Mori et al., 1983*)' or 'internal TCs (*Igarashi et al., 2012*)', those cells with somata that only partially reside in the mitral cell layer were excluded from analysis due to their ambiguous identity as MCs or TCs. GCs located in the MCL or GC layer were distinguished from other cell types and classified as sGCs or dGCs as previously described (*Burton and Urban, 2015*). Specifically, GCs were classified as dGCs if their apical dendritic gemmules were visibly concentrated in the deep half of the EPL, while GCs were classified

as sGCs if their apical dendritic gemmules were visibly concentrated in the superficial half of the EPL. Cell morphologies were reconstructed under a 100X oil-immersion objective and analyzed with Neurolucida (MBF Bioscience). Anatomical positions of GC apical dendritic gemmules were manually identified from 3D reconstructions using custom software written in Matlab (Mathworks).

## Electrophysiology

Cells were visualized using infrared differential interference contrast video microscopy. For MC and TC recordings, slices were continuously superfused with 37°C oxygenated Ringer solution that contained 0.2 mM $Mg^{2+}$ unless otherwise noted. Current clamp recordings were made from individual cells using electrodes filled with (in mM) 120 potassium gluconate, 2 KCl, 10 Hepes, 10 sodium phosphocreatine, 4 Mg-ATP, 0.3 $Na_3GTP$, 0.2 EGTA, 0–0.025 Alexa Fluor 594 (Life Technologies, Carlsbad, CA) and 0.2% Neurobiotin (Vector Labs, Burlingame, CA). Voltage clamp recordings were made using electrodes filled with (in mM): 140 Cs-gluconate, 10 HEPES, 2 KCl, 10 sodium phosphocreatine, 3 Mg-ATP, and 0.3 $Na_3GTP$.

M72 photostimulation was provided by a 250 µm multimode optical fiber (Thorlabs) coupled to a high-intensity light emitting diode (M470F1; Thorlabs) and driver (DC2100: Thorlabs) controlled by TTL pulses. For photostimulation in OMP-ChR2-YFP, slices were illuminated with 100 ms light pulses by a xenon arc lamp directed through an YFP filter set and 60x water-immersion objective centered on a single glomerulus. Photostimulation was confined to single glomeruli by closing the field stop as previously described (*Burton and Urban, 2015*). All data were low-pass filtered at 4 kHz and digitized at 10 kHz using a MultiClamp 700A amplifier (Molecular Devices, Sunnyvale, CA, USA) and an ITC-18 acquisition board (Instrutech, Mineola, NY, USA) controlled by custom software written in Igor Pro (WaveMetrics, Lake Oswego, OR, USA).

For GC recordings, slices were continuously superfused with warmed oxygenated Ringer's solution (temperature measured in bath: 32°C) containing 1 mM $Mg^{2+}$ and 2 mM $Ca^{2+}$. Current clamp recordings were made as described above. Voltage clamp recordings were made using electrodes filled with either the Cs-based solution supplemented with 10 mM QX−314 and 0.2% Neurobiotin or the K-based solution. To examine GC activity following activation of a single glomerulus, extracellular stimulation of olfactory sensory neuron fibers within a single glomerulus was performed as previously described (*Burton and Urban, 2015*).

## Data analysis

Lateral inhibitory currents were measured in 5 trials at a holding potential of +10 mV. Analysis of IPSCs was performed using custom Matlab (Mathworks) analysis software. The presence or absence of IPSCs was calculated by taking the average trace of 5 trials and finding the mean and standard deviation of the trace during the second prior to photostimulation. Then the baseline current (mean of the second prior to photostimulation) was subtracted from each trace. IPSCs evoked through lateral inhibition were present if positive deflections of the current trace exceeded 3*s.d. for longer than 10 ms in the 500 ms time window following M72 photostimulation.

Lateral inhibitory currents were split into early and late phases and the peak current amplitude and charge transfer were calculated in each. Charge transfer was calculated as the integral of the current trace in either the early phase (0–250 ms) or late phase (250–1500 ms) following photostimulation.

The effect of lateral inhibition on spiking was measured by performing FI curves in MCs and TCs via somatic current injection of increasing amplitudes. At each current step (500 ms), we measured the number of action potentials evoked with and without M72 photostimulation (10 ms pulses at 15 Hz). Two full FI curves (a full FI curve is defined as having current steps with and without photostimulation) were performed on each cell and the average change in firing rate at each current step was calculated and used for the presented analysis. Lateral inhibition was defined to have a significant effect on a cell if there was a greater than 10% decrease in firing rate in at least 2 consecutive current steps.

Intrinsic biophysical properties of GCs, including passive membrane, action potential, and spike train properties were calculated as previously described (*Burton and Urban, 2015*).

## Computational model

Models of odors and olfactory bulb circuitry were developed in Matlab (Mathworks) (see Source code). The scheme for generating odor panels and individual odor trials is depicted in *Figure 6—figure supplement 1*. Each odor is represented by the spatial pattern of a $15 \times 10$ array of pixels (i.e. glomeruli). For each odor presented at a particular concentration ($I_{presentation}$), we first made Odor 1. To do this, we randomly sampled 1/3 of pixels to represent non-activatable (NA) pixels that are not responsive to odors, regardless of concentration. Next a concentration threshold ($T_i$) was sampled from a uniform distribution between 0 and 1000 for each activatable pixel ($T_i = U([0,1000])$). Activatable pixels were then divided into ON and OFF pixels. ON pixels were defined as having $T_i < I_{presentation}$, while OFF pixels are ones where $T_i > I_{presentation}$. Each ON pixel then got a mean activation intensity sampled from a normal distribution ($I_{base} = N(I_{presentation}, 50)$).

The next step in the construction of the odor panel was to make an arbitrary number of other odors that are 90% similar to Odor 1. To make these odors, we enforced a set of rules.

1. Each pixel in Odor 1 had a 10% probability of changing.
2. For each NA pixel that was chosen to change:
   - $P(NA - NA) = 1/3$
   - $P(NA - ON) = 2/3$
   - If a NA pixel became an ON pixel, it received an activation intensity. $I_{base}=N(I_{presentation}, 50)$
3. For each ON pixel that was chosen to change:
   - $P(ON - NA) = 1/3$
   - $P(ON - ON) = 2/3$
   - if an ON pixel remained ON, it received a new activation intensity. $I_{base}=N(I_{presentation}, 50)$
4. For each OFF pixel that was chosen to change:
   - $P(OFF - OFF) = 1$

The last step in odor panel construction was to add noise to create trial-to-trial variability. To do this, we enforced a set of rules for sampling the activation strength of each pixel on each trial.

1. The strength of activation of NA and OFF pixels: $I_{trial} = U([0,1000])$
2. The strength of activation of ON pixels: $I_{trial} = N(I_{base}, 5)$

These odors became the glomerular input for olfactory bulbs composed of either MC or TCs. Each MC/TC was represented by a continuous firing rate variable, v. Each neuron received a leak current, an inhibitory current and an excitatory current input from 1 pixel (*Equation 1*). The excitatory current was calculated by passing the pixel activation strength through a sigmoid. Two differences between MCs and TCs, higher excitability and increased excitatory input are reflected by increased slope (0.007 for MCs and 0.01 for TCs) and reduced midpoint (500 for MCs and 350 for TCs) of the TC sigmoid. MCs/TCs were randomly connected to 75/60% of other MC/TCs, reflecting the shorter extent of lateral dendrites in TCs compared with MCs (*Igarashi et al., 2012*; *Burton and Urban, 2014*). Inhibition was calculated as the product of the sum of network activity, u(t) (*Equation 5*), and a Gaussian function that determines which range of firing rates are influenced by lateral inhibition (*Equations 3,4*). The center and width of the MC and TC Gaussian distribution are based on differences in the range of rates influenced by lateral inhibition in our data.

$$t\frac{dv}{dt} = stim(t) - leak(t) - inhib(t) \tag{1}$$

$$leak = g_l * (v(t) - E_l) \tag{2}$$

$$inhib_{tufted}(t) = g_i * u(t) * e^{-\frac{(v(t)-\mu t)^2}{\sigma t}}, \ \mu_t = 10, \ \sigma_t = 350 \tag{3}$$

$$inhib_{mitral}(t) = g_i * u(t) * e^{-\frac{(v(t)-\mu m)^2}{\sigma m}}, \ \mu_m = 50, \ \sigma_m = 500 \tag{4}$$

$$inhib_{subtractive}(t) = g_i * u(t) \tag{5}$$

$$\text{inhib}_{\text{divisive}}(t) = g_i * v(t) \tag{6}$$

$$u(t) = \sum_{n=1}^{T} v_n(t), \text{ where } T \text{ is the number of connected MCs or TCs} \tag{7}$$

$$stim(t) = f(I_{\text{trial}}), \text{ where } f \text{ is a sigmoid} \tag{8}$$

One of N odors from the panel was presented to the MC or TC bulb on each of 2000 trials. The MC or TC outputs from half of the trials are used to train a naïve Bayes classifier which was then used to predict which odor is being presented on the remaining half of trials. The percent correct is used as the discrimination accuracy for that odor panel. For simulations, we constructed 100 odor panels at each concentration. Concentrations in *Figures 6* are plotted as the percent of maximum concentration. The maximum concentration was defined as the concentration that evokes maximum firing rates in MCs and TCs, which in our models is set at 100 Hz – a rate often observed in vivo. A sigmoidal transfer function was used to translate glomerular inputs into MC/TC outputs so that odor concentrations could be defined in terms of MC/TC firing rates while not make any explicit comparisons to actual odor concentrations. Significance in discrimination accuracy between output neuron configurations was determined in 2 steps. First, we conducted one-way ANOVA tests (corrected for multiple comparisons – i.e. the number of odor concentrations tested) on the discrimination accuracy of the three output neuron configurations at each odor concentration. At concentrations with significant ANOVA tests, we performed post-hoc t-tests to determine whether one particular output neuron configuration was significantly better than the other two. The results of the post-hoc t-tests determine the highlighted areas of *Figure 6*.

The simulations used to generate the data in *Figure 6k–m,* in which 2 output neurons receive inputs from 1 glomerulus required additional connectivity rules. For bulbs containing only MCs or TCs, each of the two cell that input from the same glomerulus receive inhibition from a random set of other neurons. In bulbs containing both MCs and TCs, inhibition remained segregated, that is TCs/MCs only received inhibition from a random set of other TCs/MCs. Additionally, for all 3 output neuron configurations, neurons projecting to the same glomerulus did not inhibit one another. The simulations used to generate the data in *Figure 6—figure supplement 3*, in which one input image served as input to 2 separate populations that only differed in the range of ADLI. For each population, each pixel in the image provided into to one neuron.

## Acknowledgements

Thanks to Brent Doiron, Claire Cheetham and members of the Urban laboratory for helpful comments and discussion, and Greg LaRocca for excellent technical assistance. This work was supported by the National Institute of Deafness and Other Communication Disorders Grants R01-DC011184 (NNU) and F31-DC013490 (SDB).

## Additional information

### Funding

| Funder | Grant reference number | Author |
| --- | --- | --- |
| National Institute on Deafness and Other Communication Disorders | RO1-DC011184 | Nathan N Urban |
| National Institute on Deafness and Other Communication Disorders | F31-DC013490 | Shawn D Burton |

The funders had no role in study design, data collection and interpretation, or the decision to submit the work for publication.

## Author contributions

MAG, SDB, Conception and design, Acquisition of data, Analysis and interpretation of data, Drafting or revising the article, Contributed unpublished essential data or reagents; NNU, Conception and design, Analysis and interpretation of data, Drafting or revising the article, Contributed unpublished essential data or reagents

## Author ORCIDs

Shawn D Burton, http://orcid.org/0000-0002-8907-6487
Nathan N Urban, http://orcid.org/0000-0002-0365-9068

## Ethics

Animal experimentation: All experiments were completed in compliance with the guidelines established by the Institutional Animal Care and Use Committee of Carnegie Mellon University.(IACUC # AS15-010) and University of Pittsburgh (IACUC # 15116582).

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
