## [Decision Letter]

Thank you for submitting your article "Distinct lateral inhibitory circuits drive parallel processing of sensory information in the mammalian olfactory bulb" for consideration by *eLife*. Your article has been reviewed by three peer reviewers, and the evaluation has been overseen by Naoshige Uchida as the Reviewing Editor and Gary Westbrook as the Senior Editor. The reviewers have discussed the reviews with one another and the Reviewing Editor has drafted this decision to help you prepare a revised submission.

Summary:

The authors report that tufted cells are inhibited by lateral inhibition from neighboring glomeruli at lower firing rate ranges than for mitral cells. That is, activity-dependent lateral inhibition (ADLI) occurs at different firing rate ranges in tufted versus mitral cells. The authors also present evidence suggesting that the difference in ADLI is caused by differences in excitation of two sub-classes of granule cells. Finally, the authors used computational modeling to show that differential ADLIs between mitral and tufted cells help improve discrimination of odors at wider concertation ranges.

All the reviewers thought the main finding were very interesting. During discussion between the reviewers, however, there were different opinions whether it was necessary to determine whether differential ADLIs occurs in vivo. Although such an experiment would greatly strengthen the significance of this work, it is *eLife*'s policy not to suggest experiments that will take more than 2 months. Reviewers agreed that the experiments in reduced preparations (slice) have certain advantages. However, the reviewers had other concerns that were considered substantive and therefore the authors must address these concerns before publication.

Essential revisions:

1) It is not convincing that the difference in ADLIs between mitral and tufted cells is not a result of tissue slicing. Furthermore, it is unclear whether the different properties of superficial versus deep granule cells that might have caused the differences in ADLIs are a side effect of slicing. The authors have to provide more convincing, quantitative arguments to exclude artifacts associated with slicing.

2) The authors examined the significance of differential ADLI between mitral and tufted cells using modeling. However, mitral and tufted cells are different not only in their ADLI properties but also their intrinsic and synaptic input properties. In the modeling, all of these are changed at the same time, and therefore, it is unclear whether improved odor discriminability at different odor concentrations is due to ADLI alone, differences in intrinsic/synaptic properties alone, or a combination of all factors. Please clarify this issue with additional analysis.

3) In Figure 3, the authors mention that "decreases in firing rate were calculated as the average decrease across all significantly affected trials". Selecting trials based on "significantly affected trials" seems inappropriate. Exclusion of such trials in tufted cells would have resulted in an artificially large value for the percent change in spike rate due to ADLI.

4) in vivo experiments would greatly increase the significance of the present study as lateral inhibition may be sensitive to subtle details of the network properties, spontaneous activity and breathing etc. that can be seen only in an intact network. If the authors decide not to perform in vivo experiments, potential caveats of their in vitro experiments must be explicitly discussed.

*Reviewer #1:*

This study explores differences in the olfactory bulb's output mitral cells (MCs) and tufted cells (TCs) in a previously-described phenomenon known as activity dependent lateral inhibition (ADLI). Evidence is provided based on slice recordings that MCs and TCs differ in the kinetics of lateral inhibition, the spike frequency range over which ADLI is observed, as well was the relative effect of ADLI on spiking. Also, recordings from granule cells (GCs) suggest that differences in ADLI in MCs versus TCs could reflect differences in excitation of two sub-classes of GCs. Finally, computational modeling is used to make the case that differences in ADLI between MCs and TCs can confer differences in the concentration ranges over which the two neuron types best discriminate odors and may also expand the concentration ranges over which odors can be effectively discriminated by the bulb.

Overall, this is an interesting study that addresses a broadly relevant question: How can two parallel sensory processing paths (in this case, mediated by MCs and TCs) enhance sensory discrimination? Much of the study also employs a powerful experimental paradigm involving optogenetic stimulation of a single, molecularly-defined glomerulus (M72) as a means to drive lateral inhibition. This strategy should eliminate ambiguities about whether only one glomerulus is being stimulated and, also, should reduce variabilities across experiments. Some of the results are also basically convincing, for example that MCs and TCs have different kinetics for lateral inhibition and also differ in the frequency range over which ADLI occurs. However, I have a number of concerns with both the experimental and modeling sections of the paper.

1) The analysis methods used raise questions about results suggesting that TCs display a larger percent decrease in spike rate due to ADLI than MCs (Figure 3). The legend of Figure 3 indicates that changes in firing rate due to ADLI were calculated based only on "significantly affected trials". Because ADLI in TCs occurred at low spike frequencies, it seems quite possible that many of the trials in which small effects were actually occurring may have been missed by a per-trial statistical analysis (due to a low total number of spikes). Exclusion of such trials in TCs would have resulted in an artificially large value for the percent change in spike rate due to ADLI, as they calculated it. Such exclusions would be less common in MCs that displayed ADLI at higher spike frequencies.

2) That differences in the properties of superficial versus deep GCs (sGCs and dGCs) contribute to differences in ADLI between MCs and TCs is not completely convincing. The argument is based on the parallel observations that TCs display ADLI at lower spike frequencies than MCs and also that sGCs (which may preferentially contact TCs) are more responsive to glomerular stimulation. However, the authors need to exclude better the possibility that the greater spike probability and evoked currents in sGCs is not a slicing artifact. sGCs, which are closer to glomeruli, may naturally be expected to have more of their input circuitry intact in slices. The fact that sGCs appear to have greater intrinsic excitability helps their argument, but it is not sufficient. Comparing ADLI and the responsiveness of GCs is also complicated by the fact that the experiments assaying these two features appear to have been conducted in differing magnesium concentrations (0.2 mM versus 1 mM).

The authors also use the fact that the mGluR agonist DHPG reduces the frequency range for ADLI in MCs to argue that differences in ADLI between MCs and TCs are due to GC properties. This is based on prior observations that DHPG can enhance GC excitability. DHPG appears not alter MC spiking due to direct current injection (Figure 5—figure supplement 1), yet other circuit effects of DHPG need to be better controlled. For example, DHPG could enhance stimulus-evoked lateral inhibition by altering glomerular layer neurons. This, in turn, could cause the same frequency shifts in ADLI in MCs as increases in GC excitability.

3) The computational results suggesting that differences in ADLI between MCs and TCs enable these cells to perform odor discrimination at different concentrations are not completely convincing. The most important results are shown in Figure 6, which displays the output of a model that incorporates differences in the level of sensory neuron input onto TCs versus MCs, differences in intrinsic excitability, as well as differences in ADLI. This shows that TCs perform better discrimination at lower odor concentration than MCs. However, it is unclear how much differences in ADLI between MCs versus TCs are contributing to these discrimination curves. The fact that TCs discriminate better at lower concentrations may mainly reflect their greater responsiveness to low concentrations of odor. The authors need to test whether removing inhibition substantially impacts the discrimination curves in Figure 6, as well as Figure 6, as they did for their simpler model in Figure 6.

*Reviewer #2:*

Geramita, Burton & Urban report through a series of elegant experiments in acute olfactory bulb preparations that the two types of bulb principal neurons, tufted and mitral cells, receive different levels of inhibition from granule cells (stronger and more asynchronous to mitral cells compared to tufted cells). The authors show through a combination of patch-clamp recordings, current injections and optogenetic stimulation of genetically labeled glomeruli that activity dependent lateral inhibition (ADLI) occurs at low firing rate regimes in tufted cells and intermediate firing rates in mitral cells. They further propose as underlying substrate at play for the different ADLI regimes two distinct populations of superficial and deep granule cells (sGCs, dGCs). Anatomical reconstructions and clustering suggest that sGCs and dGCs contact differentially the lateral dendrites of TC and respectively MCs. Further functional analysis reveals differences in intrinsic excitability and threshold for action potential firing between sGCs and dGCs consistent with their proposed roles. In my opinion, the manuscript adds important novel understanding by documenting that functional differences between mitral and tufted cells emerge not only due to differences in intrinsic excitability, but also in their local connectivity with inhibitory interneurons. These findings bring timely mechanistic insight that parsimoniously explains recently published differences in TCs and MCs responses to odors, across concentrations, and to modulation of cortical feedback. Therefore, I consider the manuscript fit for publication in *eLife* should the authors address several concerns listed below.

Importantly, I have doubts regarding the necessity of the computational model presented and some of its underlying assumptions. Given the temporal complexity of active sampling (sniffing) in vivo, relevant time window for odor identification (200-300 ms, Uchida & Mainen, 2003), possible effects of the slice preparation on the functional connectivity of the bulb, and lack of top-down regulation (glutamatergic, GABA-ergic and neuromodulatory input) of mitral and tufted cell activity in the slice preparation, I suggest focusing the manuscript on the experimental results with fewer incursions into the simulations.

1) The authors present in Figure 6 two sigmoids that govern the responses of MC and TCs to increasing concentrations. MCs have been reported to display a wide range of concentration response curves that are not necessarily monotonic (Meredith, 1986), and thus at odds with the chosen transfer function.

2) What is the functional relevance of the concentration regime sampled? What does 20% to 80% of maximal concentration translate to in terms of absolute odor concentration? Are such concentrations ever encountered by the animal?

3) Why is the classifier trained independently for each of the concentrations sampled? This seems an unrealistic regime compared to processes that may occur in the brain.

4) Differences in functional properties of sGCs and dGCs can be in principle heavily modulated by top-down signals that are absent in the slice. The current version of the manuscript does not discuss this possibility.

5) MC and TCs have been reported to project with differential biases to downstream target areas. How do such biases contribute to the ability of the decoder to incorporated odor information carried by both TC and MC?

In Figure 1 to assess in an unbiased manner the contribution of GC-mediated vs. glomerular layer mediated lateral inhibition on MCs vs. TCs, the analysis (before vs. after drug cocktail application) should be restricted only onto those cells that have the apical dendrite intact. Otherwise, by construction, the contribution of the glomerular layer lateral inputs is reduced/absent in cells with apical dendrite cut. In the current version, 1 out of 5 MCs, and respectively 2 out of 5 TCs have their apical dendrite cut.

*Reviewer #3:*

The authors present a detailed in vitro analysis of differences in lateral inhibition experienced by MCs and TCs. They attribute these differences to different innervation of different populations of GCs and use a simplified model to illustrate that the differential inhibition can lead to improved discriminability overall (with TCs aiding to low-concentration and MCs to high-concentration discrimination).

In a previous submission (to another journal) there were numerous criticisms that the authors largely address well. One key concern that was reiterated by all reviewers was that the findings of differential inhibition were exclusively based on in vitro experiments (and simulations) with little direct link to how sensory information might be processed. Here I think the current manuscript is still lacking. While the authors argue – rightly so – that they perform an in depth mechanistic analysis (which is difficult in vivo) I have to say I tend to agree in that the relevance / role of the differential inhibition for odor processing is still unclear. If indeed – as stated in the response to prior reviewer 1 – the variability / signal-to-noise will mask the effect in vivo one might question how dominant it could be in a behaving context. I surely don't argue for repeating all experiments in an awake behaving animal but key experiments (parts of Figure 1 and Figure 4) at least in an anesthetized (if not in an awake) animal would convince the reader that the effects studied are indeed shaping odor processing in vivo.

[Editors' note: further revisions were requested prior to acceptance, as described below.]

Thank you for resubmitting your work entitled "Distinct lateral inhibitory circuits drive parallel processing of sensory information in the mammalian olfactory bulb" for further consideration at *eLife*. Your revised article has been favorably evaluated by Gary Westbrook (Senior editor), a Reviewing editor, and three reviewers.

The manuscript has been improved but there are some remaining issues that need to be addressed before acceptance as agreed upon by the reviewers in discussion following their original reviews that are appended below. Specifically, while all the referees agreed that the manuscript has been greatly improved, there are two remaining concerns. First, the reviewers thought that it remains unclear whether the authors can completely exclude the possibility that slicing biased the contribution of superficial versus deep granule cells in activity-dependent lateral inhibition (ADLI) (point # 1 of reviewer 1). Furthermore, the interpretation of the experiment using DHPG should be revised so that the possibility of the network effect (point #2, reviewer 1) is appreciated. We think that both points can be addressed with revisions to the text as outlined by reviewer 1. For more details, please see the reviewers' original reviews below.

*Reviewer #1:*

The Reviewers have adequately addressed most of my concerns. Especially helpful are their new modeling results showing that differences in ADLI between MCs and TCs contribute to odor discrimination across a wider range of odor concentrations even in the presence of other important differences between MCs and TCs. Some concerns about slice artifacts and their interpretation of differences in excitation of sGCs and dGCs however remain, as do concerns about their DHPG experiments (both brought up in Major Point 2 of my prior review). These, which are explained below, however will just require changes in the text. Their overall story is quite interesting, and I would be satisfied if they simply backed off on some of their claims pertaining to GCs.

1) That the greater evoked responses in sGCs versus dGCs is due to greater excitatory input and intrinsic excitability rather than slice artifacts is still not completely convincing. In response to this concern, raised previously, the authors provided three lines of evidence in the rebuttal letter and Results section, which I will address individually (authors' text is in parentheses).

A) (Our in vitro observation of greater sGC firing following glomerular activation (Figure 4) corresponds well with the previous in vivo observation of stronger odor-evoked activity in putative sGCs (Wellis and Scott, 1990).)

This argument is not very helpful because the stronger odor-evoked activity in sGCs in vivo could simply be due to the well-described greater odor-responsiveness of tufted cells (to which the sGCs may preferentially connect). The results do not support that sGCs have synaptic and intrinsic properties that make them more responsive.

B) (Examination of GC biophysical properties revealed several intrinsic differences supporting greater recruitment of sGCs than dGCs, including a more hyperpolarized action potential threshold in sGCs (Figure 4; Figure 4—figure supplement 2, Table 2) and greater intrinsic excitability in sGCs in response to somatic step current injections (Figure 4; Figure 4—figure supplement 2, Table 3), despite equivalent somatodendritic sizes (Figure 4—figure supplement 1; Figure 4—figure supplement 2, Table 1) and passive membrane properties (Figure 4—figure supplement 2, Table 4) between sGCs and dGCs. Collectively these observations provide an independent line of evidence that supports our conclusions about differences between sGCs and dGCs.)

These data are somewhat helpful but they do not directly address whether the differences in the evoked responses in sGCs versus dGCs are not mainly a slice artifact.

C) (Analysis of spontaneous synaptic activity revealed no difference in event frequency or amplitude between sGCs and dGCs….Critically, recordings of spontaneous synaptic activity were performed in the absence of TTX and thus contain some degree of action potential-dependent input, which likely originates from intact presynaptic cells. Therefore, equal spontaneous event frequencies between sGCs and dGCs suggests that their respective presynaptic circuits are comparably intact. Moreover, equal spontaneous event amplitudes suggest a comparable contribution of larger action potential-dependent and smaller action potential-independent events between sGCs and dGCs, again consistent with comparably intact presynaptic circuits.)

These data are not especially helpful for excluding slice artifacts because the spontaneous events in GCs (sEPSCs) could reflect spontaneous release of glutamate from pieces of M/T lateral dendrites that remain in the slice but are not associated with a cell body. This differs from responses evoked by stimulation of a single glomerulus, which requires intact cell bodies so that action potentials can pass to the lateral dendrites. That the experiments were conducted in the absence of TTX is not very supportive of there being intact cell bodies, since no evidence is provided to suggest that a major portion of the sEPSCs are being driven by action potentials. They could be mEPSCs (and, notably, the mean amplitudes of the sEPSCs are quite small). In addition, it cannot be excluded that there are spontaneous action potentials in pieces of lateral dendrites that are dissociated from cell bodies. Full-sized action potentials can propagate down mitral cell lateral dendrites, suggesting the presence of sodium channels in these dendrites. To use the spontaneous events as an argument against slice artifacts, the authors would need to perform additional controls (e.g., recordings of miniature events in TTX).

Taking these points together, I still am not convinced that the differences in evoked responses in dGCs and sGCs do not mainly reflect a slice artifact. However, at this point, I would be satisfied if the authors make changes in the text and reduce their emphasis on differences in the synaptic and intrinsic properties of the two GC sub-types being causal to the different ADLI properties of MCs versus TCs.

2) Some concern also remains about the DHPG experiments, which the authors use to argue that the requirement for higher spike frequencies in MCs to obtain ADLI reflects the excitatory properties of GCs. I previously raised the issue that the DHPG-induced shifts in the effective range of lateral inhibition could reflect drug-effects on glomerular layer neurons rather than GCs. In response, the authors point out that 2 MCs with cut apical dendrites (that do not receive inputs from glomerular layer cells) show similar DHPG-induced shifts in the effective frequency range for ADLI as those MCs with intact apical dendrites. However, these data do not exclude the following plausible network effect involving DHPG-induced excitation of external tufted cells in the glomerular layer (Dong and Ennis, 2014): DHPG-induced excitation of eTCs causes greater feedforward excitation of a population of MCs at a glomerulus; this in turn leads to greater excitation of GCs and GC-mediated inhibition. Such an effect, involving GC inputs onto MC lateral dendrites, would not depend on a test MC having intact apical dendrites.

Despite this caveat, their results showing that the 2 MCs with cut apical dendrites show similar DHPG-induced shifts in the effective range of lateral inhibition as MCs are helpful. They at least support the conclusion that changes in the degree of GC excitation (either direct or indirect) can control the frequency range over which the ADLI occurs, which is itself interesting.

*Reviewer #2:*

The authors addressed the concerns raised and, in my opinion, the manuscript is fit for publication at this point.

*Reviewer #3:*

The authors are discussing my key concern (whether the findings hold true in vivo) very well in the rebuttal and following their logic it is indeed very likely that most of the key findings will hold true. I find the well-structured and expanded (compared to the Discussion) overview given in the rebuttal very helpful and would suggest incorporating this version into the Discussion.

---

## [Author Response]

*Essential revisions:*

*1) It is not convincing that the difference in ADLIs between mitral and tufted cells is not a result of tissue slicing. Furthermore, it is unclear whether the different properties of superficial versus deep granule cells that might have caused the differences in ADLIs are a side effect of slicing. The authors have to provide more convincing, quantitative arguments to exclude artifacts associated with slicing.*

We agree with the reviewers that the issue of tissue slicing is important. However, we believe that the differences between MCs and TCs and between sGCs and dGCs reported here are not slicing artifacts, but true features of the olfactory bulb circuit. Below, we provide rationale for why each of the three main experimental findings are not artifacts of the slicing procedure. We additionally state how our revised manuscript presents these rationale.

Main finding 1: Lateral inhibitory currents are larger and more asynchronous onto MCs than TCs (Figure 1)

Due to slicing, some MCs and TCs that project to the M72 glomerulus may be truncated, and cells that reside farther from the M72 glomerulus will be expected to have a higher probability of being truncated. Therefore, we are likely to lose more M72 MCs than M72 TCs with the slicing procedure. Additionally, prior work (Mori, et al., J Comp Neurol,1983; Scott, et al., J Comp Neurol, 1983) shows that lateral dendrites of MCs and TCs project radially in all directions. Therefore, the lateral dendrites of the remaining MCs and TCs still connecting to the M72 glomerulus are likely to be affected approximately equally by the slicing procedure. Consequently, we would expect that slicing would be more likely to reduce M72-mediated lateral inhibition onto MCs than TCs. However, in Figure 1, we find that MCs have stronger lateral inhibition than TCs. Therefore, the differences in lateral inhibition that we report are likely an underestimation of the actual differences between MCs and TCs. Additionally, Figure 1—figure supplement 1 shows that the strength of lateral inhibitory currents does not vary with distance from the M72 in either MCs or TCs.

Main finding 2: Lateral inhibition affects intermediate firing rates in MCs and low firing rates in TCs (Figure 2, Figure 3)

Similar to the above argument, we include data in Figure 3—figure supplement 1 that shows that the effect of lateral inhibition on spiking does not vary with distance from the M72 glomerulus in either MCs or TCs. Additionally, we have added panels (E-J) to Figure 3—figure supplement 1 that show that neither the lower nor upper bound of the effective range of lateral inhibition depend on the distance from the M72 glomerulus in either MCs or TCs.

Finally, the lateral dendrites of MCs and TCs innervate distinct strata of the external plexiform layer and these differences could result in differences in the fraction of the original connections left intact following slicing. However, the effect of lateral inhibition on MCs that are observed here following activation of an entire glomerulus closely mirrors the effects that were observed when activating a single presynaptic MC in our previous work (Arevian et al., Nat Neuro, 2008). That is, only the probability of finding connections and the absolute magnitude of the effect are increased by activating all of the cells associated with a glomerulus (compared to the prior study when only a single presynaptic cell was activated). This suggests that the critical feature of the effect (maximal lateral inhibition at intermediate firing rates) is a property of single connections rather than one that depends on having a full repertoire of connections intact.

Main finding 3: Compared with deep GCs, superficial GCs spike with higher probability and receive stronger excitatory currents following glomerular stimulation (Figure 4)

We agree with the reviewers that the greater recruitment of sGCs following glomerular activation may be a consequence of our use of an acute slice preparation. Specifically, as TC circuitry is closer to any given glomerulus than MC circuitry, TC-mediated input to GCs (likely sGCs) may be better preserved than MC-mediated input to GCs (likely dGCs) in the acute slice, leading to stronger sGC excitation and recruitment following glomerular activation. We explicitly note this as a caveat in the Results section of our revised manuscript.

We note, however, that three lines of evidence argue against this possibility, and support greater feedforward recruitment of sGCs as a physiological feature of the olfactory bulb circuit.

A) Our in vitro observation of greater sGC firing following glomerular activation (Figure 4) corresponds well with the previous in vivoobservation of stronger odor-evoked activity in putative sGCs (Wellis and Scott, 1990). In the Results section of our revised manuscript, we now explicitly discuss this prior in vivostudy in the context of our current results.

B) Examination of GC biophysical properties revealed several intrinsic differences supporting greater recruitment of sGCs than dGCs, including a more hyperpolarized action potential threshold in sGCs (Figure 4; Figure 4—figure supplement 2, Table 2) and greater intrinsic excitability in sGCs in response to somatic step current injections (Figure 4; Figure 4—figure supplement 2, Table 3), despite equivalent somatodendritic sizes (Figure 4—figure supplement 1; Figure 4—figure supplement 2, Table 1) and passive membrane properties (Figure 4—figure supplement 2, Table 4) between sGCs and dGCs. Collectively these observations provide an independent line of evidence that supports our conclusions about differences between sGCs and dGCs.

C) Analysis of spontaneous synaptic activity revealed no difference in event frequency or amplitude between sGCs and dGCs. We have included this new analysis in Figure 4—figure supplement 2, Table 5 of the revised manuscript. Critically, recordings of spontaneous synaptic activity were performed in the absence of TTX and thus contain some degree of action potential-dependent input, which likely originates from intact presynaptic cells. Therefore, equal spontaneous event frequencies between sGCs and dGCs suggests that their respective presynaptic circuits are comparably intact. Moreover, equal spontaneous event amplitudes suggest a comparable contribution of larger action potential-dependent and smaller action potential-independent events between sGCs and dGCs, again consistent with comparably intact presynaptic circuits.

In view of these three lines of evidence, we believe that our results strongly suggest that sGCs are more strongly recruited than dGCs following activation of a single glomerulus due to stronger excitatory input and greater intrinsic excitability.

In addition to the specific revisions to our manuscript described above, we have also broadly summarized this rationale in the following section of our revised Discussion:

“Performing these experiments in acute slices provides the best opportunity to explore the causes and consequences of differences in lateral inhibition between MCs and TCs. […]Addressing the extent to which ADLI affects odor-evoked responses in MCs and TCs in vivois thus an important future direction.”

*2) The authors examined the significance of differential ADLI between mitral and tufted cells using modeling. However, mitral and tufted cells are different not only in their ADLI properties but also their intrinsic and synaptic input properties. In the modeling, all of these are changed at the same time, and therefore, it is unclear whether improved odor discriminability at different odor concentrations is due to ADLI alone, differences in intrinsic/synaptic properties alone, or a combination of all factors. Please clarify this issue with additional analysis.*

We agree with the reviewer that understanding how each of the differences between MCs and TCs contribute to concentration-dependent differences in odor discrimination is important and have updated Figure 6 to explore the contributions of these specific differences. First, in Figure 6, we show that MC-like and TC-like ADLI is alone sufficient to allow otherwise identical cells to best discriminate between high and low concentration odors, respectively. Second, in Figure 6, we show that differences in intrinsic excitability and glomerular input strength between MCs and TCs (in the absence of ADLI differences) also allow MCs and TCs to best discriminate between high and low concentration odors, respectively. Finally, we perform the simulation in which all three factors differ between MCs and TCs (Figure 6) and find that together these three factors work additively to allow MCs and TCs to best discriminate between high and low concentration odors, respectively.

We make two conclusions from these simulations. First, each of three experimentally determined differences between MCs and TCs (intrinsic excitability, glomerular input strength, and ADLI) alone is sufficient to allow TCs perform better at low concentration. Second, TCs are best at performing low concentration odor discrimination and MCs are best at performing high concentration discriminations when all three differences are present. Therefore, because these intrinsic and circuit-level differences between MCs and TCs all lead to the same functional difference between MCs and TCs, this functional difference is likely to be an important feature of processing of odor-evoked information.

We have updated the Results, Figure 6 and the Figure 6 legend, to reflect these new analyses and conclusions.

*3) In Figure 3, the authors mention that "decreases in firing rate were calculated as the average decrease across all significantly affected trials". Selecting trials based on "significantly affected trials" seems inappropriate. Exclusion of such trials in tufted cells would have resulted in an artificially large value for the percent change in spike rate due to ADLI.*

We apologize for the confusion caused by the Figure 3 legend. The statement “decreases in firing rate were calculated as the average decrease across all significantly affected trials” only refers to Figure 3, in which distinct trials corresponded to distinct firing rates. In Figure 3, data from all trials (so that none were excluded) were used, and these data demonstrate that lateral inhibition reduces the absolute firing rate by similar amounts in both MCs and TCs.

To reduce confusion, we have changed the Figure 1 legend to state, “In F, decreases in firing rate were calculated as the average decrease across all significantly affected firing rates.”

*4) in vivo experiments would greatly increase the significance of the present study as lateral inhibition may be sensitive to subtle details of the network properties, spontaneous activity and breathing etc. that can be seen only in an intact network. If the authors decide not to perform in vivo experiments, potential caveats of their in vitro experiments must be explicitly discussed.*

We agree that exploring how ADLI influences odor-evoked responses in MCs and TCs in vivowill be important to the field. However, we believe that the reduced slice preparation that we use in our work provides several advantages for exploring the novel features and mechanisms of ADLI in MCs and TCs. Therefore, we have added an additional section to our Discussion that more explicitly addresses the advantages and limitations of performing our experiments in vitro. We refer the reviewer to this section and to our response to Essential Revision #1 above.

*Reviewer #1:*

*Overall, this is an interesting study that addresses a broadly relevant question: How can two parallel sensory processing paths (in this case, mediated by MCs and TCs) enhance sensory discrimination? Much of the study also employs a powerful experimental paradigm involving optogenetic stimulation of a single, molecularly-defined glomerulus (M72) as a means to drive lateral inhibition. This strategy should eliminate ambiguities about whether only one glomerulus is being stimulated and, also, should reduce variabilities across experiments. Some of the results are also basically convincing, for example that MCs and TCs have different kinetics for lateral inhibition and also differ in the frequency range over which ADLI occurs. However, I have a number of concerns with both the experimental and modeling sections of the paper.*

*1) The analysis methods used raise questions about results suggesting that TCs display a larger percent decrease in spike rate due to ADLI than MCs (Figure 3). The legend of Figure 3 indicates that changes in firing rate due to ADLI were calculated based only on "significantly affected trials". Because ADLI in TCs occurred at low spike frequencies, it seems quite possible that many of the trials in which small effects were actually occurring may have been missed by a per-trial statistical analysis (due to a low total number of spikes). Exclusion of such trials in TCs would have resulted in an artificially large value for the percent change in spike rate due to ADLI, as they calculated it. Such exclusions would be less common in MCs that displayed ADLI at higher spike frequencies.*

We apologize for the confusion caused by the Figure 3 legend and refer the reviewer to our response to Essential Revision #3 above.

*2) That differences in the properties of superficial versus deep GCs (sGCs and dGCs) contribute to differences in ADLI between MCs and TCs is not completely convincing. The argument is based on the parallel observations that TCs display ADLI at lower spike frequencies than MCs and also that sGCs (which may preferentially contact TCs) are more responsive to glomerular stimulation. However, the authors need to exclude better the possibility that the greater spike probability and evoked currents in sGCs is not a slicing artifact. sGCs, which are closer to glomeruli, may naturally be expected to have more of their input circuitry intact in slices. The fact that sGCs appear to have greater intrinsic excitability helps their argument, but it is not sufficient. Comparing ADLI and the responsiveness of GCs is also complicated by the fact that the experiments assaying these two features appear to have been conducted in differing magnesium concentrations (0.2 mM versus 1 mM).*

We agree that understanding how the slicing may contribute to our reported findings is vital and have addressed this issue above under Essential Revision #1.

*The authors also use the fact that the mGluR agonist DHPG reduces the frequency range for ADLI in MCs to argue that differences in ADLI between MCs and TCs are due to GC properties. This is based on prior observations that DHPG can enhance GC excitability. DHPG appears not alter MC spiking due to direct current injection (Figure 5—figure supplement 1), yet other circuit effects of DHPG need to be better controlled. For example, DHPG could enhance stimulus-evoked lateral inhibition by altering glomerular layer neurons. This, in turn, could cause the same frequency shifts in ADLI in MCs as increases in GC excitability.*

The reviewer brings up a good point that DHPG could enhance lateral inhibition mediated by glomerular layer circuits. However, we have indicated in Figure 5 whether MCs have intact apical dendrites. We find that the 2 MCs with cut apical dendrites show similar shifts in the effective range of lateral inhibition as those MCs with intact apical dendrites – that DHPG decreases the range of rates influenced by lateral inhibition. Therefore, we conclude that the effects of DHPG on glomerular layer interneurons are insufficient to explain the shifts in ADLI. We have clarified this point in the Results section of our revised manuscript.

*3) The computational results suggesting that differences in ADLI between MCs and TCs enable these cells to perform odor discrimination at different concentrations are not completely convincing. The most important results are shown in Figure 6, which displays the output of a model that incorporates differences in the level of sensory neuron input onto TCs versus MCs, differences in intrinsic excitability, as well as differences in ADLI. This shows that TCs perform better discrimination at lower odor concentration than MCs. However, it is unclear how much differences in ADLI between MCs versus TCs are contributing to these discrimination curves. The fact that TCs discriminate better at lower concentrations may mainly reflect their greater responsiveness to low concentrations of odor. The authors need to test whether removing inhibition substantially impacts the discrimination curves in Figure 6, as well as Figure 6, as they did for their simpler model in Figure 6.*

We have made Figure 6 clearer and refer the reviewer to our response to Essential Revision #2 above.

*Reviewer #2:*

*Geramita, Burton & Urban report through a series of elegant experiments in acute olfactory bulb preparations that the two types of bulb principal neurons, tufted and mitral cells, receive different levels of inhibition from granule cells (stronger and more asynchronous to mitral cells compared to tufted cells). The authors show through a combination of patch-clamp recordings, current injections and optogenetic stimulation of genetically labeled glomeruli that activity dependent lateral inhibition (ADLI) occurs at low firing rate regimes in tufted cells and intermediate firing rates in mitral cells. They further propose as underlying substrate at play for the different ADLI regimes two distinct populations of superficial and deep granule cells (sGCs, dGCs). Anatomical reconstructions and clustering suggest that sGCs and dGCs contact differentially the lateral dendrites of TC and respectively MCs. Further functional analysis reveals differences in intrinsic excitability and threshold for action potential firing between sGCs and dGCs consistent with their proposed roles. In my opinion, the manuscript adds important novel understanding by documenting that functional differences between mitral and tufted cells emerge not only due to differences in intrinsic excitability, but also in their local connectivity with inhibitory interneurons. These findings bring timely mechanistic insight that parsimoniously explains recently published differences in TCs and MCs responses to odors, across concentrations, and to modulation of cortical feedback. Therefore, I consider the manuscript fit for publication in eLife should the authors address several concerns listed below.*

*Importantly, I have doubts regarding the necessity of the computational model presented and some of its underlying assumptions. Given the temporal complexity of active sampling (sniffing) in vivo, relevant time window for odor identification (200-300 ms, Uchida & Mainen, 2003), possible effects of the slice preparation on the functional connectivity of the bulb, and lack of top-down regulation (glutamatergic, GABA-ergic and neuromodulatory input) of mitral and tufted cell activity in the slice preparation, I suggest focusing the manuscript on the experimental results with fewer incursions into the simulations.*

*1) The authors present in Figure 6 two sigmoids that govern the responses of MC and TCs to increasing concentrations. MCs have been reported to display a wide range of concentration response curves that are not necessarily monotonic (Meredith, 1986), and thus at odds with the chosen transfer function.*

We thank the reviewer for bringing this study to our attention. In our updated manuscript (Results, fourteenth paragraph), we have added this reference as a caveat to our assumption that MCs and TCs encode odor concentration as firing rate changes.

While the Meredith work indicates that some MC/TCs may display concentration response curves that are not monotonic, the other cited studies show that firing rate is one of several properties that consistently vary with changes in concentration. Even if some MCs and TCs do not display monotonic concentration response curves, the results of our simulations will still be important for understanding how ADLI may influence the representations encoded in large numbers of MCs and TCs.

*2) What is the functional relevance of the concentration regime sampled? What does 20% to 80% of maximal concentration translate to in terms of absolute odor concentration? Are such concentrations ever encountered by the animal?*

The x-axis in the Figure 6 plots the “percent maximum concentration”. The maximum concentration is defined as the concentration that evokes maximum firing rates in MCs and TCs, which in our models is set at 100 Hz – a rate often observed in vivo. We use a sigmoidal transfer function to translate glomerular inputs into MC/TC outputs so that we can define the odor concentrations in terms of MC/TC firing rates while not making any explicit comparisons to actual odor concentrations. Given that odor concentration is defined in terms of MC/TC firing rates, 20-80% of maximal concentration refer to odors that evoke firing rates in MCs and TCs that are 20-80% of their maximum firing rates, which are known to occur in vivo.

We have included an explanation of our rationale for using this quantification metric in the updated Methods (subsection “Computational model”, fifth paragraph).

*3) Why is the classifier trained independently for each of the concentrations sampled? This seems an unrealistic regime compared to processes that may occur in the brain.*

The reviewer’s question is an important one. We chose to train the classifier independently for each concentration to mimic typical behavioral experiments where animals are trained to discriminate between different odors presented at similar odor concentrations. We believe that in order to train the classifier on odors presented at multiple concentrations, we would need to introduce assumptions about how the same odor is represented at different concentrations. Indeed, this question of how MC/TC representations of odor identity remain invariant across a range of concentrations is important but ultimately one that cannot be adequately addressed with our model.

In the manuscript (Results), we have added the following caveat. “While understanding how animals discriminate between odors presented at a variety of concentrations is important, we confined our discriminations to odors presented at the same concentration to more closely match behavioral experiments in mice (Abraham et al., Neuron, 2010; Lepousez et al., Neuron, 2010) and to keep from making a number of assumptions about how the representation of individual odors varies across concentration.”

*4) Differences in functional properties of sGCs and dGCs can be in principle heavily modulated by top-down signals that are absent in the slice. The current version of the manuscript does not discuss this possibility.*

We agree with the reviewer that both superficial and deep GCs may be heavily influenced by top-down modulation from a variety of brain areas and have addressed this caveat in the updated manuscript. Our response to this point as well as our discussion of other differences that may occur in vivocan be found under Essential Revisions #1 and #4 above.

*5) MC and TCs have been reported to project with differential biases to downstream target areas. How do such biases contribute to the ability of the decoder to incorporated odor information carried by both TC and MC?*

The reviewer brings up a good point that MCs and TCs project to separate sets of downstream areas. However, current data suggest that there are areas, such as the anterior piriform cortex, that receive input from both MCs and TCs. The simulation that generated the data in Figure 6 relies on a decoder that incorporates information from both MCs and TCs. This model assumes that neurons in the piriform cortex are performing some kind of clustering/categorization of both MC and TC outputs. This is a relatively common assumption and is supported by a variety of data that suggest the anterior piriform cortex is functioning as an associative memory system performing pattern completion (Haberly, Trends in Neuro, 1989; Bekkers, Trends in Neuro, 2013). In the updated manuscript (Discussion, last paragraph), we have included an explicit reference to the anterior piriform cortex as an area that might be integrating input from both MCs and TCs.

Our simulations do not address how other areas such as the anterior olfactory tubercle, which receives stronger input from TCs than MCs, might be integrating or decoding information transmitted in both channels. Therefore, we have refrained from drawing conclusions about how information is integrated in these other downstream areas.

*In Figure 1 to assess in an unbiased manner the contribution of GC-mediated vs. glomerular layer mediated lateral inhibition on MCs vs. TCs, the analysis (before vs. after drug cocktail application) should be restricted only onto those cells that have the apical dendrite intact. Otherwise, by construction, the contribution of the glomerular layer lateral inputs is reduced/absent in cells with apical dendrite cut. In the current version, 1 out of 5 MCs, and respectively 2 out of 5 TCs have their apical dendrite cut.*

We agree that in order to compare GC-mediated and glomerular layer mediated inhibition between MCs and TCs, only cells with intact apical dendrites should be analyzed. However, after removing cells with cut apical dendrites and redoing the statistical analysis done in Figure 1, we find no statistical difference between MCs and TCs. Therefore, in the revised manuscript, we have kept cells with cut apical dendrites in Figure 1 in order to emphasize the point that lateral inhibitory currents are eliminated after reducing GC-mediated inhibition. However, in the manuscript we have added, “Removing cells with cut apical dendrites and redoing the analysis in Figure 1 similarly shows that GCs contribute similar proportions of lateral inhibition onto MCs and TCs.”

*Reviewer #3:*

*The authors present a detailed in vitro analysis of differences in lateral inhibition experienced by MCs and TCs. They attribute these differences to different innervation of different populations of GCs and use a simplified model to illustrate that the differential inhibition can lead to improved discriminability overall (with TCs aiding to low-concentration and MCs to high-concentration discrimination).*

*In a previous submission (to another journal) there were numerous criticisms that the authors largely address well. One key concern that was reiterated by all reviewers was that the findings of differential inhibition were exclusively based on in vitro experiments (and simulations) with little direct link to how sensory information might be processed. Here I think the current manuscript is still lacking. While the authors argue – rightly so – that they perform an in depth mechanistic analysis (which is difficult in vivo) I have to say I tend to agree in that the relevance / role of the differential inhibition for odor processing is still unclear. If indeed – as stated in the response to prior reviewer 1 – the variability / signal-to-noise will mask the effect in vivo one might question how dominant it could be in a behaving context. I surely don't argue for repeating all experiments in an awake behaving animal but key experiments (parts of Figure 1 and Figure 4) at least in an anesthetized (if not in an awake) animal would convince the reader that the effects studied are indeed shaping odor processing in vivo.*

We thank the reviewer for his comments. We agree that understanding how lateral inhibition affects odor-evoked responses in MCs and TCs in vivois critical. We have added a paragraph to our Discussion that addresses the limitations of our in vitroapproach. It can be found above under Essential Revision #1.

[Editors' note: further revisions were requested prior to acceptance, as described below.]

*The manuscript has been improved but there are some remaining issues that need to be addressed before acceptance as agreed upon by the reviewers in discussion following their original reviews that are appended below. Specifically, while all the referees agreed that the manuscript has been greatly improved, there are two remaining concerns. First, the reviewers thought that it remains unclear whether the authors can completely exclude the possibility that slicing biased the contribution of superficial versus deep granule cells in activity-dependent lateral inhibition (ADLI) (point # 1 of reviewer 1). Furthermore, the interpretation of the experiment using DHPG should be revised so that the possibility of the network effect (point #2, reviewer 1) is appreciated. We think that both points can be addressed with revisions to the text as outlined by reviewer 1. For more details, please see the reviewers' original reviews below.*

*Reviewer #1:*

*The Reviewers have adequately addressed most of my concerns. Especially helpful are their new modeling results showing that differences in ADLI between MCs and TCs contribute to odor discrimination across a wider range of odor concentrations even in the presence of other important differences between MCs and TCs. Some concerns about slice artifacts and their interpretation of differences in excitation of sGCs and dGCs however remain, as do concerns about their DHPG experiments (both brought up in Major Point 2 of my prior review). These, which are explained below, however will just require changes in the text. Their overall story is quite interesting, and I would be satisfied if they simply backed off on some of their claims pertaining to GCs.*

*1) That the greater evoked responses in sGCs versus dGCs is due to greater excitatory input and intrinsic excitability rather than slice artifacts is still not completely convincing. In response to this concern, raised previously, the authors provided three lines of evidence in the rebuttal letter and Results section, which I will address individually (authors' text is in parentheses).*

*A) (Our* in vitro *observation of greater sGC firing following glomerular activation (Figure 4) corresponds well with the previous in vivo observation of stronger odor-evoked activity in putative sGCs (Wellis and Scott, 1990).)*

*This argument is not very helpful because the stronger odor-evoked activity in sGCs in vivo could simply be due to the well-described greater odor-responsiveness of tufted cells (to which the sGCs may preferentially connect). The results do not support that sGCs have synaptic and intrinsic properties that make them more responsive.*

*B) (Examination of GC biophysical properties revealed several intrinsic differences supporting greater recruitment of sGCs than dGCs, including a more hyperpolarized action potential threshold in sGCs (Figure 4; Figure 4—figure supplement 2, Table 2) and greater intrinsic excitability in sGCs in response to somatic step current injections (Figure 4; Figure 4—figure supplement 2, Table 3), despite equivalent somatodendritic sizes (Figure 4—figure supplement 1; Figure 4—figure supplement 2, Table 1) and passive membrane properties (Figure 4—figure supplement 2, Table 4) between sGCs and dGCs. Collectively these observations provide an independent line of evidence that supports our conclusions about differences between sGCs and dGCs.)*

*These data are somewhat helpful but they do not directly address whether the differences in the evoked responses in sGCs versus dGCs are not mainly a slice artifact.*

*C) (Analysis of spontaneous synaptic activity revealed no difference in event frequency or amplitude between sGCs and dGCs….Critically, recordings of spontaneous synaptic activity were performed in the absence of TTX and thus contain some degree of action potential-dependent input, which likely originates from intact presynaptic cells. Therefore, equal spontaneous event frequencies between sGCs and dGCs suggests that their respective presynaptic circuits are comparably intact. Moreover, equal spontaneous event amplitudes suggest a comparable contribution of larger action potential-dependent and smaller action potential-independent events between sGCs and dGCs, again consistent with comparably intact presynaptic circuits.)*

*These data are not especially helpful for excluding slice artifacts because the spontaneous events in GCs (sEPSCs) could reflect spontaneous release of glutamate from pieces of M/T lateral dendrites that remain in the slice but are not associated with a cell body. This differs from responses evoked by stimulation of a single glomerulus, which requires intact cell bodies so that action potentials can pass to the lateral dendrites. That the experiments were conducted in the absence of TTX is not very supportive of there being intact cell bodies, since no evidence is provided to suggest that a major portion of the sEPSCs are being driven by action potentials. They could be mEPSCs (and, notably, the mean amplitudes of the sEPSCs are quite small). In addition, it cannot be excluded that there are spontaneous action potentials in pieces of lateral dendrites that are dissociated from cell bodies. Full-sized action potentials can propagate down mitral cell lateral dendrites, suggesting the presence of sodium channels in these dendrites. To use the spontaneous events as an argument against slice artifacts, the authors would need to perform additional controls (e.g., recordings of miniature events in TTX).*

*Taking these points together, I still am not convinced that the differences in evoked responses in dGCs and sGCs do not mainly reflect a slice artifact. However, at this point, I would be satisfied if the authors make changes in the text and reduce their emphasis on differences in the synaptic and intrinsic properties of the two GC sub-types being causal to the different ADLI properties of MCs versus TCs.*

We thank the reviewer for these comments and agree that we cannot completely rule out the possibility that the differences between evoked responses in sGCs and dGCs may be due to slice effects. We also agree that we have not provided direct evidence that differences between dGCs and sGCs cause differences in ADLI between MCs and TCs. However, we believe that we do provide evidence that differences in ADLI are not caused of glomerular layer lateral inhibitory circuits and are thus due to external plexiform layer circuits. Additionally, the data presented in Figure 4 and Figure 5 constitute two independent lines of indirect evidence that support the hypothesis that differences in GCs are one mechanism responsible for the observed differences in ADLI between MCs and TCs. However, we cannot rule out the possibility that other mechanisms also contribute to the observed differences in ADLI.

The revised manuscript reflects this shift in tone in several areas listed below:

A) “Several previous findings are consistent with this hypothesis. GC soma position correlates with GC subtype…”

B) “However, despite the fact that these observations were made in vitro, three lines of evidence support greater feedforward recruitment of sGCs as a physiological feature of the olfactory bulb circuit. First, our in vitroobservation of…”

C) “In total, our results thus support the hypothesis that sGCs are more strongly recruited than dGCs following activation of a single glomerulus due to stronger excitatory input and greater intrinsic excitability, although we cannot rule out an important role for other mechanisms, especially in the intact system in vivo.”

D) “As an additional test of our hypothesis that differences in the excitability of GCs provide one potential mechanism underlying thefunctional difference in ADLI between MCs and TCs, we measured the effects of lateral inhibition on MC spiking before and after increasing the excitability of GCs with the mGluR agonist, DHPG.”

E) “Despite these observations, however, we note that increased GC excitability may not be the sole mechanism responsible for these DHPG-induced shifts in the effective range of lateral inhibition. For instance, DHPG-induced increases in the excitability of external tufted(Dong and Ennis 2014) cells may, in turn, increase the activity in M72-MCs, the activity in GCs and GC-mediated inhibition onto the recorded MC.”

F) “These data (Figure 4–Figure 5) provide two indirect lines of evidence that suggest that differences in GC populations provide one mechanism for differences in the effective activity range of lateral inhibition onto MCs and TCs.”

G) “We provide evidence that these effects could be caused, in part, by differences in the excitability of subclasses of GCs that preferentially inhibit TCs vs. MCs.”

H) “Additionally, we provide indirect evidence that differences in the recruitability of sGCs and dGCs may be one potential mechanism underlying of the differences in ADLI between MCs and TCs. Our data suggests that sGCs spike with a higher probability than dGCs following the activation of a single glomerulus.”

I) “Additionally, these differences between sGCs and dGCs can explain why the upper bound on the effective activity range of lateral inhibition differs between MCs and TCs.”

*2) Some concern also remains about the DHPG experiments, which the authors use to argue that the requirement for higher spike frequencies in MCs to obtain ADLI reflects the excitatory properties of GCs. I previously raised the issue that the DHPG-induced shifts in the effective range of lateral inhibition could reflect drug-effects on glomerular layer neurons rather than GCs. In response, the authors point out that 2 MCs with cut apical dendrites (that do not receive inputs from glomerular layer cells) show similar DHPG-induced shifts in the effective frequency range for ADLI as those MCs with intact apical dendrites. However, these data do not exclude the following plausible network effect involving DHPG-induced excitation of external tufted cells in the glomerular layer (Dong and Ennis, 2014): DHPG-induced excitation of eTCs causes greater feedforward excitation of a population of MCs at a glomerulus; this in turn leads to greater excitation of GCs and GC-mediated inhibition. Such an effect, involving GC inputs onto MC lateral dendrites, would not depend on a test MC having intact apical dendrites.*

*Despite this caveat, their results showing that the 2 MCs with cut apical dendrites show similar DHPG-induced shifts in the effective range of lateral inhibition as MCs are helpful. They at least support the conclusion that changes in the degree of GC excitation (either direct or indirect) can control the frequency range over which the ADLI occurs, which is itself interesting.*

We agree with the reviewer that DHPG-induced shifts in the effective range of lateral inhibition onto MCs can, in part, be explained by increases in GC excitability, based on the 2 MCs that lack apical dendrites. However, we agree that we cannot completely exclude the possibility that DHPG also increases the excitability of ETCs, which, in turn, can increase the activity in M72 MCs, the activity of GCs, and the amount of GC-mediated inhibition.

Therefore, in the updated manuscript we have added the following sentence: “As a caveat, however, we note that increased GC excitability may not be the sole mechanism responsible for these DHPG-induced shifts in the effective range of lateral inhibition. For instance, DHPG-induced increases in the excitability of external tufted cells (Dong and Ennis, 2014) may, in turn, increase the activity in M72-MCs, the activity in GCs and GC-mediated inhibition onto the recorded MC.”

*Reviewer #3:*

*The authors are discussing my key concern (whether the findings hold true in vivo) very well in the rebuttal and following their logic it is indeed very likely that most of the key findings will hold true. I find the well-structured and expanded (compared to the Discussion) overview given in the rebuttal very helpful and would suggest incorporating this version into the Discussion.*

We thank the reviewer and agree that it is useful to add more of the detailed logic from our first response to reviewers. In addition to the changes above, several of which address this issue and reiterate the logic from the previous rebuttal letter, we have added the following to our Discussion:

“Additionally, the lateral dendrites of MCs and TCs innervate distinct strata of the external plexiform layer and these differences could result in differences in the fraction of original connections left intact following slicing. […] This suggests that the critical feature of the effect (maximal lateral inhibition at intermediate firing rates) is a property of single connections rather than one that depends on having a full repertoire of connections intact.”